# Kolmogorov-Arnold Networks for Time Series Granger Causality Inference

## Abstract

We propose the Granger causality inference Kolmogorov-Arnold Networks (KANGCI), a novel architecture that extends the recently proposed Kolmogorov-Arnold Networks (KAN) to the domain of causal inference. By extracting base weights from KAN layers and incorporating the sparsity-inducing penalty and ridge regularization, KANGCI effectively infers the Granger causality from time series. Additionally, we propose an algorithm based on time-reversed Granger causality that automatically selects causal relationships with better inference performance from original/time-reversed time series or integrates the results to improve performance. Comprehensive experiments on Lorenz-96, Gene regulatory networks, fMRI BOLD signals, VAR, and real-world EEG datasets demonstrate that the proposed model achieves competitive performance to state-of-the-art methods in inferring Granger causality from nonlinear, high-dimensional, and limited-sample time series.

## 1 Introduction

Granger causality is a statistical framework for analyzing the causal relationship between time series. It offers a powerful tool to investigate temporal dependencies and the direction of influence between variables (Shojaie & Fox, 2022). By examining the past values of time series, Granger causality seeks to determine if the historical knowledge of one variable improves the prediction of another (Bressler & Seth, 2011). Granger causality has been used in many fields, such as econometrics (Mele et al., 2022), neuroscience (Chen et al., 2023), climate science (Ren et al., 2023), etc.

Recently, there has been a growing interest in incorporating the neural network into the study of Granger causality due to its inherent nonlinear mapping capabilities. For now, a variety of neural Granger causality models have been proposed, mainly based on multi-layer perceptron (MLP) (Tank et al., 2022; Bussmann et al., 2021; Zhou et al., 2024), recurrent neural network (RNN) (Khanna & Tan, 2019; Tank et al., 2022), convolutional neural network (CNN) (Nauta et al., 2019), or their combination (Cheng et al., 2024). These models have achieved significant improvements in inferring nonlinear Granger causality but still have some limitations: (1) RNN-based models are more suitable for processing long time series but experience decreased inference performance in the limited time-sample scenario. (2) MLP-based models face the challenge of low inference efficiency when dealing with high-dimensional and noisy time series. (3) CNN-based models perform ineffectively on many nonlinear datasets.

Therefore, our motivation is to propose a neural network-based Granger causality model that can effectively infer causal relationships from high-dimensional nonlinear time series with limited sampling points. We consider a novel framework, the Kolmogorov-Arnold Network (KAN) (Liu et al., 2024), to construct a Granger causality inference model. The rationale for choosing KAN arises from two aspects. First, KAN leverages the Kolmogorov–Arnold representation theorem and employs learnable splines to approximate functions, which enables it to flexibly capture complex and even non-smooth patterns that are difficult for traditional MLPs with fixed activation functions (Liu et al., 2024). Second, KAN's architecture favors multiplication over division during function approximation, thereby avoiding numerical instability when divisors approach zero and ensuring smoother gradients and more reliable optimization, which further enhances its robustness in real-world applications (Somvanshi et al., 2024).

Our work extends the basic KAN to the field of causal inference and aims to evaluate whether the KAN-based model has the potential to outperform MLP-based, RNN-based, and CNN-based baselines. Our main contributions are as follows:

- We propose a simple but effective Granger causality model based on KAN. The model only needs to extract base weights of KAN layers and impose the sparsity-inducing penalty and ridge regularization to infer Granger causality.
- We propose a time-reversed Granger causality (TRGC) algorithm that automatically selects causal relationships with the higher inference performance from origin/time-reversed time series or further improves performance by fusing both of them.
- Extensive experiments on Lorenz-96, Gene regulatory networks, fMRI BOLD, VAR, and real-world EEG datasets validate that the proposed model attains stable and competitive performances in Granger causality inference.

## 2 BACKGROUND

### 2.1 COMPONENT-WISE NONLINEAR AUTOREGRESSIVE (NAR)

Assume a $p$-dimensional time series with $T$ samples in each demension. In the nonlinear autoregressive (NAR) model, the $t^{th}$ time point $x_t$ can be denoted as a function $g$ of its past time values:

$$x_t = g\left(x_{<t1}, \ldots, x_{<tp}\right) + e_t \tag{1}$$

where $x_{<ti} = (x_{<(t-K)i}, \ldots, x_{<(t-2)i}, x_{<(t-1)i})$ is $K$ past values of series $i$, the function $g$ usually takes the form of the neural network or other nonlinear function, $e_t$ is the combined effect of all instantaneous and exogenous factors influencing the measurement.

Different from the NAR model that assumes the prediction of each time series depends on the same past time lag of all the series, the component-wise NAR assumes that the $t^{th}$ time point of each time series $x_{ti}$ may depend on different past time lags of all the series:

$$x_{ti} = g_i\left(x_{<t1}, \ldots, x_{<tp}\right) + e_{ti} \tag{2}$$

To infer Granger causality from the component-wise NAR model, sparsity-inducing penalty is applied:

$$\min_W \sum_{t=K}^{T} \left(x_{ti} - g_i\left(x_{<t1}, \ldots, x_{<tp}\right)\right)^2 + \lambda \sum_{j=1}^{p} \Theta\left(W_{:,j}\right) \tag{3}$$

where $W$ is extracted from the neural network, $\Theta$ is the sparsity-inducing penalty that penalizes the parameters in $W$ to zero, $\lambda$ is the hyperparameter that controls the strength of the penalty. In the NAR model, if there exists a time lag $k$, $W_{:,j}^k$ contains non-zero parameters, time series $j$ Granger-causes to time series $i$.

### 2.2 TIME REVERSED GRANGER CAUSALITY

The TRGC is initially introduced by Haufe et al. (2013), which is used to reduce spurious connections caused by volume conduction effects in analyzing Electroencephalogram (EEG) signals (van den Broek et al., 1998; Nunez et al., 1997). Subsequently, Winkler et al. (2016) demonstrate that, in finite-order autoregressive processes, causal relationships would reversed in time-reversed time series. Moreover, comparing the causal relationships inferred from the original and time-reversed time series can enhance the robustness of causal inference against noise. However, the findings of Winkler et al. (2016) primarily apply to linear systems. Recent research indicates that in nonlinear chaotic systems, causal relationships inferred from time-reversed time series generally align with those from the original data, with perfect causal relationship reversal occurring only under specific conditions (Kořenek & Hlinka, 2021).

### 2.3 KOLMOGOROV–ARNOLD NETWORKS (KAN)

Liu et al. (2024) propose KAN, which has garnered attention as a compelling alternative to MLP. The theoretical foundation of MLP is rooted in the universal approximation theorem, which demonstrates

that neural networks can approximate any continuous function under appropriate conditions (Pinkus, 1999). By contrast, KAN is grounded in the Kolmogorov-Arnold (KA) representation theorem, which states that any multivariate continuous function can be represented by the sum of a finite number of univariate functions (Schmidt-Hieber, 2021).

**Theorem 1** *Let $f : [0,1]^n \rightarrow \mathbb{R}$ be a continuous multivariate function. There exist continuous univariate functions $\Phi_q$ and $\phi_{q,p}$ such that:*

$$f(x_1, x_2, \ldots, x_n) = \sum_{q=1}^{2n+1} \Phi_q \left( \sum_{p=1}^{n} \phi_{q,p}(x_p) \right)$$

*where $\Phi_i : \mathbb{R} \rightarrow \mathbb{R}$ and $\phi_{q,p} : [0,1] \rightarrow \mathbb{R}$ are continuous functions.*

Although the KA representation theorem is elegant and general, its application in deep learning remains limited before the work of Liu et al. (2024). This limitation can be attributed to two primary factors: (1) the function $\phi_{q,p}$ is typically non-smooth. (2) the theorem is constrained to construct shallow neural networks with two-layer nonlinear architectures with limited hidden layer size. Liu et al. (2024) do not strictly constrain the neural network to fully adhere to Theorem 1, but instead extend the network to arbitrary width and depth, making it applicable for deep learning. Due to this alteration, KAN and its variants have been extensively applied across various domains, including computer vision (Bodner et al., 2024), time series forecasting (Xu et al., 2024), health informatics (Li et al., 2024).

## 3 MODEL ARCHITECTURE

### 3.1 COMPONENT-WISE KAN

To extract the influence from input to output, we model each component $g_i$ using a separate KAN. Let $g_i$ take the form of a KAN with $L-1$ layers, and $h^l$ are denoted as the $l^{th}$ hidden layer. The trainable parameter of KAN including base weight $W_{base}$ and spline weight $W_{spline}$ on each layer, where $W_{base} = \{W_b^0, W_b^1, \ldots, W_b^{L-1}\}$ and $W_{spline} = \{W_s^0, W_s^1, \ldots, W_s^{L-1}\}$. We separate the $W_{base}$ into the first layer weighs $W_b^0$, and the other layers $W_b^l$ ($0 < l < L$). By using these notations, the vector of the hidden units in the first layer $h^1$ is denoted as:

$$\mathbf{h}^1 = \underbrace{\begin{pmatrix} \phi_{0,1,1}(\cdot) & \cdots & \phi_{0,1,n_0}(\cdot) \\ \phi_{0,2,1}(\cdot) & \cdots & \phi_{0,2,n_0}(\cdot) \\ \vdots & \vdots & \\ \phi_{0,n_1,1}(\cdot) & \cdots & \phi_{0,n_1,n_0}(\cdot) \end{pmatrix}}_{\mathbf{\Phi}_0} \mathbf{x}_t \tag{4}$$

where $n_0$ is the input time series dimension, $n_1$ is the first hidden layer size. Here, the $\phi(x)$ is denoted as:

$$\phi(x) = W_b^0 \cdot x + W_s^0 \cdot \sum_i c_i B_i(x) \tag{5}$$

where $B_i$ is denoted as B-splines, $c_i$ is the control coefficients. Subsequently, The vector of the hidden units in the layer $l$ is denoted as:

$$\mathbf{h}^l = \underbrace{\begin{pmatrix} \phi_{l-1,1,1}(\cdot) & \cdots & \phi_{l-1,1,n_{l-1}}(\cdot) \\ \phi_{l-1,2,1}(\cdot) & \cdots & \phi_{l-1,2,n_{l-1}}(\cdot) \\ \vdots & \vdots & \\ \phi_{l-1,n_l,1}(\cdot) & \cdots & \phi_{l-1,n_l,n_{l-1}}(\cdot) \end{pmatrix}}_{\mathbf{\Phi}_{l-1}} \mathbf{h}^{l-1} \tag{6}$$

where $n_l$ and $n_{l-1}$ are the $l^{th}$ and $l-1^{th}$ hidden layer size, respectively. Here, the $\phi(x)$ is denoted as:

$$\phi(x) = W_b^{l-1} \cdot x + W_s^{l-1} \cdot \sum_i c_i B_i(x) \tag{7}$$

The time series $x_t$ go through the $L-1$ hidden layers to generate the output $x_{ti}$, which is denoted as:

$$x_{ti} = g_i(x_t) + e_{ti} = \Phi_{L-1} \circ h^{L-1} + e_{ti} \tag{8}$$

## 3.2 APPLYING SPARSITY-INDUCING PENALTY AND RIDGE REGULARIZATION ON KAN TO INFER GRANGER CAUSALITY

According to Eq.3, the inference of Granger causality in Eq.8 uses component-wise NAR combined with sparsity-inducing penalty. In our study, we extract the base weight of the first hidden layer $W_b^0$ and apply group lasso penalty, which is denoted as:

$$GroupLasso(W_{b(:,j)}^0) = \sum_{j=1}^{p} \left\| W_{b(:,j)}^0 \right\|_F \tag{9}$$

where $W_{b(:,j)}^0$ is the $j$ column of the $W_b^0$ corresponding to the time series $j$. $\| \cdot \|_F$ is denoted as the Frobenius matrix norm. The sparsity-inducing loss $\mathcal{L}_s$ is defined as:

$$\mathcal{L}_s = \lambda \sum_{j=1}^{p} \|W_{b(:,j)}^0\|_F \tag{10}$$

$\lambda > 0$ is the group lasso hyperparameter that controls the penalty strength. For the base weight of other hidden layers $W_b^l$, we apply ridge regularization to them, which is denoted as:

$$RidgeRegularization(W_b^{1:L-1}) = \sum_{l=1}^{L-1} \|W_b^l\|_2 \tag{11}$$

where $\| \cdot \|_2$ is denoted as the $L2$ norm. The ridge regularization loss $\mathcal{L}_r$ is defined as:

$$\mathcal{L}_r = \gamma \sum_{l=1}^{L-1} \|W_b^l\|_2 \tag{12}$$

$\gamma > 0$ is the ridge regularization hyperparameter that controls the regularization strength. Finally, the predicted loss is defined as:

$$\mathcal{L}_p = \sum_{i=1}^{p} (x_{ti} - g_i(x_t))^2 \tag{13}$$

Therefore, the loss function is defined as:

$$\mathcal{L} = \mathcal{L}_p + \mathcal{L}_s + \mathcal{L}_r \tag{14}$$

Since the proposed model is a component-wise architecture, a total of $p$ models are needed to construct the complete Granger causality matrix. We extract the first hidden layer weight $W_b^0$ to compute the $i^{th}$ row of the Granger causality matrix $G$, which is denoted as:

$$G_{(i,:)} = \|W_{b(:,j)}^0\|_F \tag{15}$$

## 3.3 FUSION OF ORIGIN AND TIME-REVERSED TIME SERIES

The TRGC is first proved by Winkler et al. (2016) in bivariate linear systems, demonstrating that integrating both the original and time-reversed time series can effectively reduce the impact of autocovariance, thereby improving the performance of causal inference. However, in nonlinear systems, the causal relationship obtained from the time-reversed time series does not exhibit the same property in linear systems. As a result, the applicability of TRGC is largely confined to the linear system, and its potential in multivariate nonlinear time series remains insufficiently investigated. Consequently, our objective is to develop a TRGC algorithm for multivariate nonlinear time series that further enhance the inference performance.

Algorithm 1 summarizes the proposed TRGC algorithm. Firstly, we need to determine whether the time series is an ideal linear AR process. The determination procedure consists of three statistical

tests. Augmented Dickey-Fuller (ADF) test for evaluating stationary, Brock-Dechert-Scheinkman (BDS) test for assessing nonlinearity, and Ljung-Box test (LB) for verifying whether the residuals from the AR model fitting are white noise. If a multivariate time series passes all the tests, we consider it an ideal AR process. For this case, only $p$ models are applied to the original time series to compute the Granger causality matrix. For other cases, a total of $2p$ KANGCI models are required, with the first $p$ models applied to the original time series and the next $p$ models to the time-reversed time series. Then, we use Eq.15 to calculate the Granger causality matrix for the original and reversed time series, respectively. Subsequently, we compare the losses to determine whether to select a single matrix or fuse both matrices. Specifically, when the prediction loss and sparsity-inducing loss of the original time series are lower than those of the reversed time series, it indicates that the model performs better in prediction and sparsity on the original time series. Therefore, the Granger causality inferred from the original time series is chosen as the final result. Conversely, if the model exhibits lower losses on time-reversed time series, the Granger causality inferred from time-reversed time series is selected. When the prediction loss and sparsity-inducing loss do not align between the two time series, we average two matrices. In our experiments, this straightforward strategy can effectively improve the Granger causality inference performance. The ablation studies of the proposed TRGC algorithm are provided in Section 4.5 and Appendix B.

---

**Algorithm 1** TRGC algorithm for inferring Granger causality with KANGCI

---

1: **Input:** The origin multivariate time series $\{x_t\}$ with dimension $p$; group lasso penalty hyperparameter $\lambda$; ridge regularization hyperparameter $\gamma$.
2: **Output:** The inferred Granger causality adjacency matrix $G$.
3: Using ADF, BDS and LB tests to determine whether the input time series is an ideal AR process.

4: **if** True **then**
5:      Train $p$ models with hyperparameter $\lambda$ and $\gamma$ and compute GC matrix $G$ on the origin time series using Eq.15.
6: **else**
7:      Let $\{\tilde{x}_t\}$ be the time-reversed time series of $\{x_t\}$, $\{x_1, x_2, \ldots, x_T\} \equiv \{\tilde{x_T}, \tilde{x_{T-1}}, \ldots, \tilde{x_1}\}$.
8:      Train first $p$ models on $\{x_t\}$ and next $p$ models on $\{\tilde{x}_t\}$) using hyperparameter $\lambda$ and $\gamma$.
9:      Compute GC matrix $G_o$ and $G_r$ from origin and time-reversed time series using Eq.15.
10:      Get predict loss $\mathcal{L}_{p(o)}$, $\mathcal{L}_{p(r)}$, sparsity-inducing loss $\mathcal{L}_{s(o)}$, $\mathcal{L}_{s(r)}$ from origin and time-reversed time series, respectively.
11:      **if** $\mathcal{L}_{p(o)} < \mathcal{L}_{p(r)}$ **AND** $\mathcal{L}_{s(o)} < \mathcal{L}_{s(r)}$ **then**
12:          $G = G_o$
13:      **else if** $\mathcal{L}_{p(o)} > \mathcal{L}_{p(r)}$ **AND** $\mathcal{L}_{s(o)} > \mathcal{L}_{s(r)}$ **then**
14:          $G = G_r$
15:      **else**
16:          $G = \frac{1}{2}(G_o + G_r)$
17:      **end if**
18: **end if**
19: **return** $G$

---

## 4 EXPERIMENT

In this section, we present the performance of KANGCI on four widely used benchmark datasets: Lorenz-96, Gene regulatory networks, fMRI BOLD signals, and VAR. Comparative experiments are conducted against several state-of-the-art models, including cMLP & cLSTM (Tank et al., 2022), TCDF (Nauta et al., 2019), eSRU (Khanna & Tan, 2019), GVAR (Marcinkevičs & Vogt, 2021), NAVAR (MLP) & NAVAR (LSTM) (Bussmann et al., 2021), CUTS+ (Cheng et al., 2024), JGC (Suryadi et al., 2023), and JRNGC (Zhou et al., 2024). Additional comparisons with other basis function-based models are provided in Appendix G. Moreover, we conduct experiments on real-world EEG signals to validate the effectiveness of KANGCI in practical applications. The corresponding results are provided in Appendix C.

In alignment with prior studies, the model performances are evaluated using the area under the receiver operating characteristic curve (AUROC) and Area Under the Precision-Recall Curve

(AUPRC). Notably, in the evaluation of the Gene regulatory networks, only the off-diagonal elements of the Granger causality adjacency matrix are considered since the gold standard provided by the Gene regulatory networks does not account for self-causality. For the Lorenz-96, fMRI BOLD, and VAR datasets, all elements of the adjacency matrix are included.

## 4.1 LORENZ-96

Lorenz-96 is a mathematical model employed to investigate the dynamics of simplified atmospheric systems. Its behavior is governed by the following ordinary differential equation:

$$\frac{\partial x_{t,i}}{\partial t} = -x_{t,i-1}\left(x_{t,i-2} - x_{t,i+1}\right) - x_{t,i} + F \tag{16}$$

where $F$ represents the external forcing term in the system. The sequence index $i$ is taken modulo, $x_{-1} = x_{p-1}$, $x_0 = x_p$, $i = 1, 2, \cdots, p$, $p$ denotes the spatial dimension of the system. The increase in $F$ results in heightened system chaos, while the increase in $p$ enhances the spatial complexity of the system. We simulate $R = 5$ replicates under the following three conditions : (1) $F = 10$, $p = 10$, $T = 1000$ (low dimensionality, weak nonlinearity); (2) $F = 40$, $p = 40$, $T = 1000$ (high dimensionality, strong nonlinearity); (3) $F = 40$, $p = 40$, $T = 500$ (limited observations).

Table 1: AUROC and AUPRC of the Lorenz-96 dataset.

| Models | AUROC | | | AUPRC | | |
|---|---|---|---|---|---|---|
| | $p = 10, F = 10$ $T = 1000$ | $p = 40, F = 40$ $T = 1000$ | $p = 40, F = 40$ $T = 500$ | $p = 10, F = 10$ $T = 1000$ | $p = 40, F = 40$ $T = 1000$ | $p = 40, F = 40$ $T = 500$ |
| cMLP | $0.983_{\pm 0.003}$ | $0.967_{\pm 0.025}$ | $0.924_{\pm 0.033}$ | $0.968_{\pm 0.002}$ | $0.931_{\pm 0.014}$ | $0.855_{\pm 0.051}$ |
| cLSTM | $0.978_{\pm 0.004}$ | $0.943_{\pm 0.027}$ | $0.863_{\pm 0.044}$ | $0.964_{\pm 0.004}$ | $0.865_{\pm 0.015}$ | $0.726_{\pm 0.035}$ |
| TCDF | $0.879_{\pm 0.011}$ | $0.763_{\pm 0.039}$ | $0.565_{\pm 0.041}$ | $0.732_{\pm 0.012}$ | $0.624_{\pm 0.042}$ | $0.545_{\pm 0.122}$ |
| eSRU | $\mathbf{1.000}_{\pm 0.000}$ | $0.973_{\pm 0.012}$ | $0.953_{\pm 0.025}$ | $\mathbf{1.000}_{\pm 0.000}$ | $0.943_{\pm 0.007}$ | $0.893_{\pm 0.023}$ |
| GVAR | $\mathbf{1.000}_{\pm 0.000}$ | $0.951_{\pm 0.016}$ | $0.941_{\pm 0.022}$ | $\mathbf{1.000}_{\pm 0.000}$ | $0.925_{\pm 0.009}$ | $0.886_{\pm 0.036}$ |
| NAVAR (MLP) | $0.993_{\pm 0.004}$ | $0.843_{\pm 0.033}$ | $0.787_{\pm 0.054}$ | $0.989_{\pm 0.005}$ | $0.742_{\pm 0.041}$ | $0.631_{\pm 0.079}$ |
| NAVAR (LSTM) | $0.993_{\pm 0.006}$ | $0.821_{\pm 0.045}$ | $0.791_{\pm 0.056}$ | $0.991_{\pm 0.005}$ | $0.784_{\pm 0.037}$ | $0.682_{\pm 0.071}$ |
| JGC | $0.994_{\pm 0.005}$ | $0.944_{\pm 0.037}$ | $0.927_{\pm 0.053}$ | $0.987_{\pm 0.004}$ | $0.923_{\pm 0.029}$ | $0.843_{\pm 0.044}$ |
| CUTS+ | $\mathbf{1.000}_{\pm 0.000}$ | $0.989_{\pm 0.003}$ | $0.961_{\pm 0.012}$ | $\mathbf{1.000}_{\pm 0.000}$ | $0.979_{\pm 0.003}$ | $0.925_{\pm 0.024}$ |
| JRNGC | $\mathbf{1.000}_{\pm 0.000}$ | $0.979_{\pm 0.012}$ | $0.956_{\pm 0.023}$ | $\mathbf{1.000}_{\pm 0.000}$ | $0.966_{\pm 0.006}$ | $0.912_{\pm 0.035}$ |
| **KANGCI** | $\mathbf{1.000}_{\pm 0.000}$ | $\mathbf{0.995}_{\pm 0.002}$ | $\mathbf{0.972}_{\pm 0.014}$ | $\mathbf{1.000}_{\pm 0.000}$ | $\mathbf{0.990}_{\pm 0.003}$ | $\mathbf{0.953}_{\pm 0.021}$ |

Table 1 presents the Granger causality inference performance of each model under three conditions. For the scenario where $p = 10$, $F = 10$, and $T = 1000$, all methods, except for TCDF, effectively infer the causal relationships. KANGCI, eSRU, GVAR, CUTS+, and JRNGC achieve AUROC and AUPRC of 1.0. However, when $p = 40$, $F = 40$, causal inference becomes more challenging, particularly as the time series length decreases. Under these conditions, the performance of cMLP, cLSTM, and NAVAR declines significantly. KANGCI achieves the highest AUROC and AUPRC. In summary, KANGCI exhibits superior performance on the Lorenz-96 dataset.

## 4.2 GENE REGULATORY NETWORKS

### 4.2.1 DREAM-3

The second dataset is the DREAM-3 in Silico Network Challenge. It consists of five sub-datasets: two corresponding to E.coli (*E.coli-1*, *E.coli-2*) and three to Yeast (*Yeast-1*, *Yeast-2*, *Yeast-3*). Each sub-dataset has a distinct ground-truth Granger causality network and includes $p = 100$ time series, representing the expression levels of $n = 100$ genes. Each time series comprises 46 replicates, sampled at 21 time points, yielding a total of 966 observations.

The AUROC of the Dream-3 dataset are shown in Table 2, AUPRC results are provided in Appendix Table 12. The performance of all models drops significantly compared to the Lorenz-96 dataset since the Dream-3 dataset contains 100 channels and carries additional noise, which leads to frequent overfitting of the models. Our model emerges as the top-performance model among its counterparts in three out of five sub-datasets. Specifically, the AUROC of the KANGCI in *E.coli-1*, *E.coli-2*, *Yeast-1* are 0.758, 0.680 and 0.667, respectively. This further proves the effectiveness of our method in identifying sparse Granger causality in high-dimensional, noisy time series.

### 4.2.2 DREAM-4

The third dataset is the DREAM-4 in silico challenge. Analogous to the DREAM-3 dataset, it consists of five sub-datasets, each containing $p = 100$ time series. However, each time series in DREAM-4 only includes 10 replicates sampled at 21 time points, yielding a total of 210 observations. This is substantially fewer than the 966 observations provided by the DREAM-3 dataset. Therefore, Dream-4 dataset challenges the inference performance of each model in scenarios with a limited number of time series observations.

Table 3 shows the improved performance of KANGCI in inferring gene-gene interactions from limited time-series data, outperforming baseline models. Specifically, our model achieves the highest AUROCs in four of the five gene networks, with values of 0.747, 0.602, 0.613, and 0.601 for networks 1, 3, 4, and 5, respectively. The AUPRC results of Dream-4 are provided in Appendix Table 13.

Table 2: AUROC of the Dream-3, T=966, p=100

| Models | AUROC | | | | |
|---|---|---|---|---|---|
| | Ecoli-1 | Ecoli-2 | Yeast-1 | Yeast-2 | Yeast-3 |
| cMLP | 0.648 | 0.568 | 0.585 | 0.511 | 0.531 |
| cLSTM | 0.651 | 0.609 | 0.579 | 0.524 | 0.552 |
| TCDF | 0.615 | 0.621 | 0.581 | 0.537 | 0.525 |
| eSRU | 0.660 | 0.636 | 0.631 | 0.561 | 0.559 |
| GVAR | 0.652 | 0.634 | 0.623 | 0.570 | 0.554 |
| NAVAR (MLP) | 0.557 | 0.577 | 0.652 | 0.573 | 0.548 |
| NAVAR (LSTM) | 0.544 | 0.473 | 0.497 | 0.477 | 0.466 |
| JGC | 0.528 | 0.536 | 0.611 | 0.558 | 0.531 |
| CUTS+ | 0.703 | 0.669 | 0.643 | 0.581 | 0.554 |
| JRNGC | 0.666 | 0.678 | 0.650 | **0.597** | **0.560** |
| **KANGCI** | **0.758** | **0.680** | **0.667** | 0.552 | 0.547 |

Table 3: AUROC of the Dream-4, T=210, p=100

| Models | AUROC | | | | |
|---|---|---|---|---|---|
| | Gene-1 | Gene-2 | Gene-3 | Gene-4 | Gene-5 |
| cMLP | 0.652 | 0.522 | 0.509 | 0.516 | 0.522 |
| cLSTM | 0.633 | 0.509 | 0.498 | 0.527 | 0.539 |
| TCDF | 0.598 | 0.491 | 0.467 | 0.567 | 0.565 |
| eSRU | 0.647 | 0.554 | 0.545 | 0.556 | 0.541 |
| GVAR | 0.662 | 0.569 | 0.565 | 0.578 | 0.542 |
| NAVAR (MLP) | 0.591 | 0.522 | 0.507 | 0.543 | 0.537 |
| NAVAR (LSTM) | 0.587 | 0.514 | 0.525 | 0.537 | 0.531 |
| JGC | 0.546 | 0.502 | 0.513 | 0.505 | 0.517 |
| CUTS+ | 0.738 | **0.622** | 0.591 | 0.584 | 0.594 |
| JRNGC | 0.731 | 0.613 | 0.583 | 0.605 | 0.580 |
| **KANGCI** | **0.747** | 0.591 | **0.602** | **0.613** | **0.601** |

### 4.3 FMRI BOLD SIGNALS

The fourth dataset is the simulated fMRI BOLD signals generated using the dynamic causal model (DCM) with the nonlinear balloon model for vascular dynamics. Each data includes multiple time series corresponding to different brain regions of interest (ROIs). Notably, the fMRI BOLD dataset contains 28 sub-datasets, each comprising 50 subjects and including distinct features. However, previous studies have typically utilized few subjects from few simulations (e.g., sim-3, sim-4) for model evaluation, which is inadequate for comprehensively assessing model performance on the fMRI dataset. In this study, we address this limitation by conducting a thorough evaluation using all subjects from all simulations (a total of 1,400 subjects). Table 4 presents the AUROC comparison results of all simulations. AUPRC results are provided in Appendix Table 11. The simulations' specifications are provided in the Appendix I Table 21.

Comparative experiments conducted on the fMRI BOLD dataset demonstrate that only TCDF, JGC, JRNGC, CUTS+, and KANGCI effectively infer Granger causality across all simulations and subjects. Among these methods, KANGCI achieved superior performance in 22 out of 28 simulations, covering various complex scenarios such as global mean confound, mixed time series, shared inputs, backward connections, cyclic connections, and time lags. In contrast, JRNGC and CUTS+ exhibited better performance in simulations with varying connection strengths (e.g., sim 15, 22, 23). Furthermore, given the inclusion of noise and randomness (with a standard deviation of 0.5 seconds in the hemodynamic response function delay) and the limited sampling points ($T = 200$) in most cases, the proposed model can more effectively infer Granger causality under noisy and data-constrained conditions compared to existing baseline models.

### 4.4 VAR

The fifth dataset is the VAR model. For a $p$-dimensional time series $x_t$, the VAR model is given by:

$$x_t = A^{(1)}x_{t-1} + A^{(2)}x_{t-2} + \ldots, + A^{(k)}x_{t-k} + u_t \tag{17}$$

where $(A^{(1)}, A^{(2)}, \ldots, A^{(k)})$ are regression coefficients matrices and $u_t$ is a vector of errors with Gaussian distribution. We define $sparsity$ as the percentage of non-zero coefficients in $A^{(i)}$, and

Table 4: AUROC of the fMRI BOLD signals, Subject=50, T=50/100/200/2000/5000

| Dateset | AUROC | | | | | | | | | | |
|---|---|---|---|---|---|---|---|---|---|---|---|
| | cMLP | cLSTM | TCDF | eSRU | GVAR | NAVAR (MLP) | NAVAR (LSTM) | JGC | CUTS+ | JRNGC | **KANGCI** |
| Sim1 | $0.746_{\pm0.04}$ | $0.689_{\pm0.05}$ | $0.806_{\pm0.03}$ | $0.729_{\pm0.04}$ | $0.753_{\pm0.05}$ | $0.723_{\pm0.05}$ | $0.711_{\pm0.05}$ | $0.812_{\pm0.05}$ | $0.825_{\pm0.04}$ | $\mathbf{0.829}_{\pm0.04}$ | $0.815_{\pm0.08}$ |
| Sim2 | $0.733_{\pm0.05}$ | $0.739_{\pm0.04}$ | $0.823_{\pm0.04}$ | $0.756_{\pm0.04}$ | $0.723_{\pm0.04}$ | $0.701_{\pm0.03}$ | $0.694_{\pm0.03}$ | $0.842_{\pm0.02}$ | $0.851_{\pm0.03}$ | $0.833_{\pm0.03}$ | $\mathbf{0.857}_{\pm0.03}$ |
| Sim3 | $0.705_{\pm0.04}$ | $0.735_{\pm0.06}$ | $0.823_{\pm0.03}$ | $0.737_{\pm0.04}$ | $0.744_{\pm0.05}$ | $0.703_{\pm0.03}$ | $0.679_{\pm0.04}$ | $0.866_{\pm0.02}$ | $0.859_{\pm0.02}$ | $0.831_{\pm0.03}$ | $\mathbf{0.884}_{\pm0.02}$ |
| Sim4 | $0.685_{\pm0.06}$ | $0.711_{\pm0.05}$ | $0.814_{\pm0.04}$ | $0.722_{\pm0.04}$ | $0.738_{\pm0.04}$ | $0.688_{\pm0.04}$ | $0.647_{\pm0.05}$ | $0.854_{\pm0.02}$ | $0.869_{\pm0.02}$ | $0.877_{\pm0.01}$ | $\mathbf{0.916}_{\pm0.01}$ |
| Sim5 | $0.681_{\pm0.05}$ | $0.691_{\pm0.04}$ | $0.815_{\pm0.03}$ | $0.756_{\pm0.04}$ | $0.732_{\pm0.03}$ | $0.794_{\pm0.03}$ | $0.812_{\pm0.04}$ | $0.838_{\pm0.03}$ | $0.849_{\pm0.04}$ | $0.851_{\pm0.05}$ | $\mathbf{0.861}_{\pm0.05}$ |
| Sim6 | $0.723_{\pm0.15}$ | $0.738_{\pm0.09}$ | $0.811_{\pm0.02}$ | $0.751_{\pm0.03}$ | $0.775_{\pm0.03}$ | $0.826_{\pm0.03}$ | $0.842_{\pm0.03}$ | $0.881_{\pm0.03}$ | $0.903_{\pm0.03}$ | $0.891_{\pm0.03}$ | $\mathbf{0.928}_{\pm0.02}$ |
| Sim7 | $0.708_{\pm0.05}$ | $0.721_{\pm0.04}$ | $0.809_{\pm0.03}$ | $0.781_{\pm0.04}$ | $0.744_{\pm0.03}$ | $0.805_{\pm0.03}$ | $0.827_{\pm0.03}$ | $0.843_{\pm0.03}$ | $0.866_{\pm0.05}$ | $0.841_{\pm0.04}$ | $\mathbf{0.902}_{\pm0.04}$ |
| Sim8 | $0.549_{\pm0.15}$ | $0.522_{\pm0.09}$ | $0.661_{\pm0.08}$ | $0.605_{\pm0.09}$ | $0.644_{\pm0.07}$ | $0.601_{\pm0.12}$ | $0.572_{\pm0.11}$ | $0.629_{\pm0.06}$ | $0.684_{\pm0.08}$ | $0.712_{\pm0.07}$ | $\mathbf{0.766}_{\pm0.08}$ |
| Sim9 | $0.667_{\pm0.07}$ | $0.704_{\pm0.09}$ | $0.789_{\pm0.05}$ | $0.710_{\pm0.05}$ | $0.679_{\pm0.06}$ | $0.713_{\pm0.08}$ | $0.727_{\pm0.08}$ | $0.752_{\pm0.07}$ | $0.819_{\pm0.06}$ | $0.806_{\pm0.06}$ | $\mathbf{0.830}_{\pm0.08}$ |
| Sim10 | $0.632_{\pm0.07}$ | $0.648_{\pm0.09}$ | $0.749_{\pm0.06}$ | $0.677_{\pm0.11}$ | $0.688_{\pm0.08}$ | $0.709_{\pm0.11}$ | $0.736_{\pm0.12}$ | $0.675_{\pm0.08}$ | $\mathbf{0.799}_{\pm0.07}$ | $0.774_{\pm0.08}$ | $0.783_{\pm0.07}$ |
| Sim11 | $0.726_{\pm0.04}$ | $0.715_{\pm0.03}$ | $0.785_{\pm0.03}$ | $0.737_{\pm0.04}$ | $0.742_{\pm0.03}$ | $0.777_{\pm0.03}$ | $0.784_{\pm0.03}$ | $0.811_{\pm0.03}$ | $0.816_{\pm0.02}$ | $0.829_{\pm0.03}$ | $\mathbf{0.837}_{\pm0.03}$ |
| Sim12 | $0.738_{\pm0.05}$ | $0.751_{\pm0.03}$ | $0.803_{\pm0.04}$ | $0.755_{\pm0.03}$ | $0.734_{\pm0.04}$ | $0.796_{\pm0.03}$ | $0.782_{\pm0.03}$ | $0.802_{\pm0.05}$ | $0.817_{\pm0.04}$ | $0.832_{\pm0.04}$ | $\mathbf{0.860}_{\pm0.03}$ |
| Sim13 | $0.596_{\pm0.07}$ | $0.586_{\pm0.04}$ | $0.714_{\pm0.06}$ | $0.655_{\pm0.08}$ | $0.676_{\pm0.09}$ | $0.685_{\pm0.08}$ | $0.693_{\pm0.09}$ | $0.683_{\pm0.09}$ | $0.716_{\pm0.07}$ | $0.739_{\pm0.07}$ | $\mathbf{0.757}_{\pm0.08}$ |
| Sim14 | $0.617_{\pm0.08}$ | $0.654_{\pm0.07}$ | $0.722_{\pm0.06}$ | $0.689_{\pm0.07}$ | $0.673_{\pm0.09}$ | $0.716_{\pm0.08}$ | $0.724_{\pm0.07}$ | $0.741_{\pm0.06}$ | $0.759_{\pm0.07}$ | $0.761_{\pm0.06}$ | $\mathbf{0.801}_{\pm0.04}$ |
| Sim15 | $0.637_{\pm0.10}$ | $0.647_{\pm0.09}$ | $0.687_{\pm0.06}$ | $0.614_{\pm0.09}$ | $0.606_{\pm0.08}$ | $0.664_{\pm0.07}$ | $0.672_{\pm0.09}$ | $0.692_{\pm0.08}$ | $0.732_{\pm0.08}$ | $\mathbf{0.773}_{\pm0.09}$ | $0.745_{\pm0.08}$ |
| Sim16 | $0.604_{\pm0.11}$ | $0.618_{\pm0.13}$ | $0.706_{\pm0.08}$ | $0.653_{\pm0.09}$ | $0.635_{\pm0.09}$ | $0.623_{\pm0.07}$ | $0.646_{\pm0.09}$ | $0.638_{\pm0.12}$ | $0.729_{\pm0.09}$ | $0.713_{\pm0.11}$ | $\mathbf{0.758}_{\pm0.09}$ |
| Sim17 | $0.694_{\pm0.05}$ | $0.686_{\pm0.05}$ | $0.813_{\pm0.03}$ | $0.712_{\pm0.04}$ | $0.704_{\pm0.05}$ | $0.769_{\pm0.03}$ | $0.781_{\pm0.04}$ | $0.794_{\pm0.04}$ | $0.845_{\pm0.03}$ | $0.862_{\pm0.04}$ | $\mathbf{0.894}_{\pm0.03}$ |
| Sim18 | $0.657_{\pm0.07}$ | $0.660_{\pm0.07}$ | $0.778_{\pm0.03}$ | $0.684_{\pm0.05}$ | $0.691_{\pm0.06}$ | $0.725_{\pm0.06}$ | $0.748_{\pm0.05}$ | $0.751_{\pm0.06}$ | $0.831_{\pm0.05}$ | $\mathbf{0.837}_{\pm0.05}$ | $0.818_{\pm0.06}$ |
| Sim19 | $0.733_{\pm0.05}$ | $0.772_{\pm0.04}$ | $0.849_{\pm0.03}$ | $0.793_{\pm0.05}$ | $0.739_{\pm0.06}$ | $0.779_{\pm0.04}$ | $0.826_{\pm0.04}$ | $0.847_{\pm0.04}$ | $0.871_{\pm0.03}$ | $0.865_{\pm0.03}$ | $\mathbf{0.906}_{\pm0.03}$ |
| Sim20 | $0.750_{\pm0.04}$ | $0.795_{\pm0.09}$ | $0.861_{\pm0.02}$ | $0.822_{\pm0.03}$ | $0.765_{\pm0.05}$ | $0.819_{\pm0.03}$ | $0.853_{\pm0.04}$ | $0.877_{\pm0.02}$ | $0.915_{\pm0.03}$ | $0.898_{\pm0.02}$ | $\mathbf{0.921}_{\pm0.03}$ |
| Sim21 | $0.651_{\pm0.07}$ | $0.674_{\pm0.08}$ | $0.753_{\pm0.05}$ | $0.707_{\pm0.06}$ | $0.719_{\pm0.04}$ | $0.688_{\pm0.05}$ | $0.702_{\pm0.06}$ | $0.643_{\pm0.08}$ | $0.786_{\pm0.06}$ | $0.767_{\pm0.06}$ | $\mathbf{0.812}_{\pm0.07}$ |
| Sim22 | $0.674_{\pm0.06}$ | $0.682_{\pm0.06}$ | $0.746_{\pm0.05}$ | $0.718_{\pm0.07}$ | $0.726_{\pm0.05}$ | $0.649_{\pm0.07}$ | $0.674_{\pm0.06}$ | $0.661_{\pm0.07}$ | $0.797_{\pm0.05}$ | $0.801_{\pm0.06}$ | $\mathbf{0.825}_{\pm0.06}$ |
| Sim23 | $0.574_{\pm0.08}$ | $0.598_{\pm0.09}$ | $0.662_{\pm0.05}$ | $0.619_{\pm0.08}$ | $0.624_{\pm0.09}$ | $0.585_{\pm0.09}$ | $0.592_{\pm0.08}$ | $0.624_{\pm0.09}$ | $0.641_{\pm0.08}$ | $\mathbf{0.705}_{\pm0.09}$ | $0.671_{\pm0.08}$ |
| Sim24 | $0.526_{\pm0.09}$ | $0.547_{\pm0.13}$ | $0.570_{\pm0.04}$ | $0.558_{\pm0.06}$ | $0.561_{\pm0.08}$ | $0.529_{\pm0.11}$ | $0.548_{\pm0.12}$ | $0.534_{\pm0.07}$ | $\mathbf{0.611}_{\pm0.07}$ | $0.581_{\pm0.07}$ | $0.594_{\pm0.09}$ |
| Sim25 | $0.627_{\pm0.07}$ | $0.613_{\pm0.05}$ | $0.681_{\pm0.04}$ | $0.633_{\pm0.07}$ | $0.641_{\pm0.05}$ | $0.608_{\pm0.06}$ | $0.595_{\pm0.07}$ | $0.645_{\pm0.04}$ | $0.707_{\pm0.06}$ | $0.728_{\pm0.06}$ | $\mathbf{0.763}_{\pm0.08}$ |
| Sim26 | $0.593_{\pm0.07}$ | $0.588_{\pm0.07}$ | $0.668_{\pm0.07}$ | $0.612_{\pm0.06}$ | $0.633_{\pm0.07}$ | $0.590_{\pm0.06}$ | $0.563_{\pm0.06}$ | $0.634_{\pm0.05}$ | $0.682_{\pm0.06}$ | $0.701_{\pm0.07}$ | $\mathbf{0.721}_{\pm0.09}$ |
| Sim27 | $0.642_{\pm0.08}$ | $0.631_{\pm0.06}$ | $0.699_{\pm0.05}$ | $0.644_{\pm0.09}$ | $0.695_{\pm0.06}$ | $0.626_{\pm0.07}$ | $0.598_{\pm0.09}$ | $0.656_{\pm0.08}$ | $0.708_{\pm0.07}$ | $0.727_{\pm0.06}$ | $\mathbf{0.753}_{\pm0.08}$ |
| Sim28 | $0.688_{\pm0.06}$ | $0.658_{\pm0.05}$ | $0.762_{\pm0.04}$ | $0.709_{\pm0.06}$ | $0.735_{\pm0.05}$ | $0.641_{\pm0.04}$ | $0.603_{\pm0.04}$ | $0.743_{\pm0.07}$ | $0.764_{\pm0.08}$ | $0.772_{\pm0.06}$ | $\mathbf{0.821}_{\pm0.07}$ |

different $sparsity$ represent different quantities of Granger causality interaction in the VAR model. The comparison results of the VAR dataset are presented in Table 5.

The comparison results reveal that all models, with the exception of TCDF, effectively infer Granger causality from the VAR dataset. Among these, CUTS+ demonstrates the highest performance, achieving an AUROC of 1.0 in three scenarios. KANGCI, JRNGC, JGC, GVAR, and eSRU achieve an AUROC of 1.0 in two scenarios. For cMLP and cLSTM, the performance decreases slightly when lag or sparsity are varied.

Table 5: AUROC of the VAR dataset.

| Models | AUROC | | | AUPRC | | |
|---|---|---|---|---|---|---|
| | $p=10, T=1000$ $sparsity=0.2$ $lag=3$ | $p=10, T=1000$ $sparsity=0.3$ $lag=3$ | $p=10, T=1000$ $sparsity=0.2$ $lag=5$ | $p=10, T=1000$ $sparsity=0.2$ $lag=3$ | $p=10, T=1000$ $sparsity=0.3$ $lag=3$ | $p=10, T=1000$ $sparsity=0.2$ $lag=5$ |
| cMLP | $\mathbf{1.0}_{\pm0.00}$ | $0.947_{\pm0.004}$ | $0.986_{\pm0.002}$ | $\mathbf{1.0}_{\pm0.00}$ | $0.832_{\pm0.006}$ | $0.973_{\pm0.004}$ |
| cLSTM | $0.986_{\pm0.004}$ | $0.921_{\pm0.004}$ | $0.961_{\pm0.003}$ | $0.964_{\pm0.005}$ | $0.854_{\pm0.006}$ | $0.911_{\pm0.006}$ |
| TCDF | $0.929_{\pm0.011}$ | $0.859_{\pm0.007}$ | $0.873_{\pm0.006}$ | $0.882_{\pm0.013}$ | $0.781_{\pm0.012}$ | $0.834_{\pm0.012}$ |
| eSRU | $\mathbf{1.0}_{\pm0.00}$ | $0.995_{\pm0.001}$ | $\mathbf{1.0}_{\pm0.00}$ | $\mathbf{1.0}_{\pm0.00}$ | $0.989_{\pm0.002}$ | $\mathbf{1.0}_{\pm0.00}$ |
| GVAR | $\mathbf{1.0}_{\pm0.00}$ | $0.992_{\pm0.002}$ | $\mathbf{1.0}_{\pm0.00}$ | $\mathbf{1.0}_{\pm0.00}$ | $0.985_{\pm0.003}$ | $\mathbf{1.0}_{\pm0.00}$ |
| NAVAR (MLP) | $0.993_{\pm0.002}$ | $0.986_{\pm0.003}$ | $0.992_{\pm0.002}$ | $0.987_{\pm0.003}$ | $0.943_{\pm0.006}$ | $0.981_{\pm0.004}$ |
| NAVAR (LSTM) | $0.993_{\pm0.002}$ | $0.963_{\pm0.004}$ | $0.987_{\pm0.002}$ | $0.985_{\pm0.003}$ | $0.957_{\pm0.007}$ | $0.959_{\pm0.003}$ |
| JGC | $\mathbf{1.0}_{\pm0.00}$ | $0.995_{\pm0.002}$ | $\mathbf{1.0}_{\pm0.00}$ | $\mathbf{1.0}_{\pm0.00}$ | $0.990_{\pm0.002}$ | $\mathbf{1.0}_{\pm0.00}$ |
| CUTS+ | $\mathbf{1.0}_{\pm0.00}$ | $\mathbf{1.0}_{\pm0.00}$ | $\mathbf{1.0}_{\pm0.00}$ | $\mathbf{1.0}_{\pm0.00}$ | $\mathbf{1.0}_{\pm0.00}$ | $\mathbf{1.0}_{\pm0.00}$ |
| JRNGC | $\mathbf{1.0}_{\pm0.00}$ | $0.997_{\pm0.001}$ | $\mathbf{1.0}_{\pm0.00}$ | $\mathbf{1.0}_{\pm0.00}$ | $0.992_{\pm0.002}$ | $\mathbf{1.0}_{\pm0.00}$ |
| **KANGCI** | $\mathbf{1.0}_{\pm0.00}$ | $0.993_{\pm0.003}$ | $\mathbf{1.0}_{\pm0.00}$ | $\mathbf{1.0}_{\pm0.00}$ | $0.987_{\pm0.002}$ | $\mathbf{1.0}_{\pm0.00}$ |

## 4.5 ABLATION STUDIES

The ablation studies are conducted to evaluate the individual contributions of each component within our model to overall performance. By selectively deactivating certain features, we derive insights into the impact of the model's fundamental elements on Granger causality inference. These analyses help to understand the model's robustness and sensitivity across varying configurations.

Four sub-datasets are selected for the model ablation: Lorenz-96 ($F=40$, $p=40$, $T=1000$), fMRI (sim-4), Dream-3 (Ecoli-1), Dream-4 (Gene-1). We evaluate the effects of omitting the proposed TRGC algorithm and changing the number of hidden units on the model performance. Moreover, a more comprehensive ablation study are presented in Appendix B.

The results are demonstrated in Table 6, 7. The proposed TRGC algorithm are proved beneficial for enhancing the performance of Granger causality inference. The advantage of tuning hidden layer units is marginal for the Lorenz-96 dataset but particularly advantageous for the fMRI, Dream-3,

Table 6: Ablation study comparing the efficacy of the proposed TRGC algorithm.

| Models | AUROC | | |
|---|---|---|---|
| | Origin | Time-Reversed | Fusion |
| Lorenz-96 | 0.995 | 0.974 | 0.995 |
| fMRI BOLD | 0.882 | 0.891 | 0.916 (↑) |
| Dream-3 | 0.728 | 0.706 | 0.758 (↑) |
| Dream-4 | 0.726 | 0.637 | 0.747 (↑) |

Table 7: Ablation study comparing the performance of KANGCI with different number of hidden units $H$.

| Models | AUROC | | | |
|---|---|---|---|---|
| | Lorenz-96 | fMRI | Dream-3 | Dream-4 |
| $H = 10$ | 0.990 | 0.831 | 0.686 | 0.692 |
| $H = 20$ | 0.995 | 0.867 | 0.683 | 0.698 |
| $H = 50$ | 0.995 | 0.882 | 0.726 | 0.719 |
| $H = 128$ | 0.988 | 0.916 | 0.758 | 0.747 |

and Dream-4 datasets. These findings confirm the efficacy of KANGCI across different scenarios and underscore the importance of tailoring the methodology.

### 4.6 MODEL COMPLEXITY

The proposed KANGCI is a component-wise model, which is similar to cMLP, cLSTM, e-SRU, etc. These models require training a total of $p$ neural networks, one for each time series. Moreover, since KANGCI utilizes time-reversed Granger causality, a total of $2p$ models are required. The comparison results on the Lorenz-96 dataset (p=40, F=40, T=1000), including AUROC, AUPRC, and the number of tunable parameters, are presented in Fig. 1.

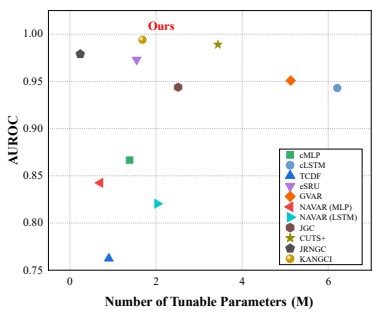 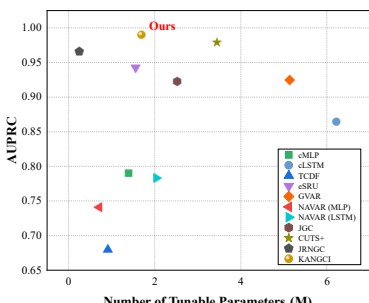

Figure 1: Performance comparisons on a 40-dimensional Lorenz-96 dataset: AUROC, AUPRC, and the number of tunable parameters.

Although KANGCI incorporates $2p$ models, the overall number of tunable parameters remains relatively modest due to its simple architecture. Compared to JRNGC, eSRU, KANGCI trades off increased time and memory consumption for improved inference performance. In contrast to models such as CUTS+ and GVAR, KANGCI not only enhances performance but also reduces computational overhead.

## 5 CONCLUSION

In this study, we propose a novel neural network-based Granger causality model, termed Granger Causality Inference Kolmogorov-Arnold Networks (KANGCI). The model leverages the base weights of KAN layers, incorporating sparsity-inducing penalty and ridge regularization to infer the causal relationship. In addition, we develop an algorithm grounded in time-reversed Granger causality to further enhance inference performances. Extensive experiments on Lorenz-96, Gene regulatory networks, fMRI BOLD, VAR, and real-world EEG signals validate that KANGCI can effectively infer Granger causality relationships from time series, outperforming the existing baselines. These results suggest that KANGCI brings a new avenue for Granger causality inference. We anticipate that this model will inspire subsequent research to design more accurate and computationally efficient frameworks for causal inference.

ETHICS STATEMENT

This research involves only simulated or publicly available benchmark datasets and does not include human or animal subjects. No ethical concerns are raised regarding privacy, fairness, or potential harmful applications.

REPRODUCIBILITY STATEMENT

Codes are provided in the supplementary materials to replicate the empirical results.

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

## A  RELATED WORKS: NEURAL NETWORK-BASED GRANGER CAUSALITY

Inferring Granger causality from nonlinear time series via neural networks has attracted widespread attention. Tank et al. (2022) proposed the cMLP and cLSTM, which extracted the first-layer weights of MLP and long short-term memory (LSTM) and imposed the sparsity-inducing penalty to infer Granger causality. Bussmann et al. (2021) proposed the Neural Additive Vector Autoregression (NAVAR) model based on MLP and LSTM, called NAVAR(MLP) and NAVAR(LSTM), for Granger causality inference. Khanna & Tan (2019) proposed the economy-SRU (eSRU) model, which extracted weights from statistical recurrent units (SRU) and also imposed sparsity-inducing penalty to infer Granger causality. Nauta et al. (2019) proposed the Temporal Causal Discovery Framework (TCDF) based on temporal convolutional network (TCN) and causal verification algorithm to infer Granger causality and select time lags. Cheng et al. (2023) proposed Causal discovery from irregUlar Time-Series (CUTS), which could effectively infer Granger causality from time series with random missing or non-uniform sampling frequency. Subsequently, to solve the problems of large causal graphs and redundant data prediction modules of CUTS, Cheng et al. (2024) proposed the CUTS+, which introduced a coarse-to-fine causal discovery mechanism and a message-passing graph neural network (MPGNN) to achieve more accurate causal reasoning. Marcinkevičs & Vogt (2021) proposed the generalised vector autoregression (GVAR) based on the self-explaining neural network model, which effectively inferred causal relationships and improved the interpretability of the model. Zhou et al. (2024) proposed a neural Granger causality model based on Jacobi regularization (JRNGC), which only needs to construct a single model for all variables to achieve causal inference. Lin et al. (2024) proposed GC-KAN, a causal inference model that leverages proximal gradient descent in conjunction with the basis functions $\phi(x)$ of KAN to capture causal relationships.

## B  FURTHER ABLATION STUDIES

### B.1  ABLATION STUDIES ON 1400 FMRI BOLD SUBJECTS

In Section 4.5, we conduct a series of ablation experiments to demonstrate the performance of the proposed TRGC algorithm. In addition, considering that the fMRI BOLD dataset has a total of 1400 subjects, statistically testing whether the model significantly improves the causal inference performance may better reflect the ability of the proposed TRGC algorithm.

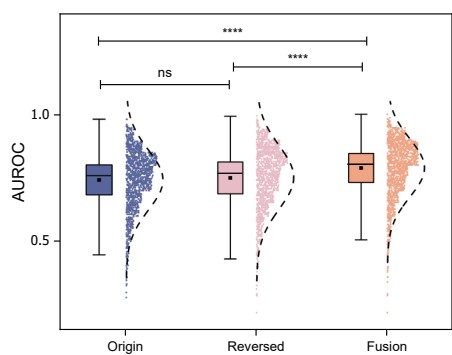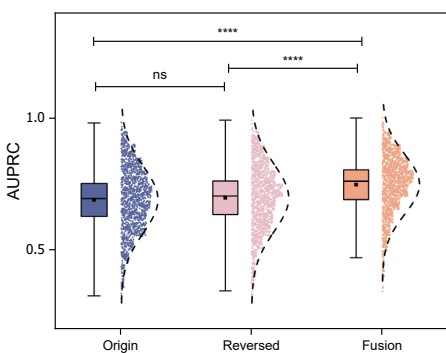

Figure 2: Ablation study comparing the efficacy of the proposed TRGC algorithm on fMRI BOLD dataset. (Left) AUROC. (Right) AUPRC. One-way ANOVA is conducted to compare the differences among Origin, Reversed, and Fusion groups, with the Tukey test for multiple comparisons.

As illustrated in Fig.2, one-way ANOVA is conducted to evaluate the between-group differences in AUROC and AUPRC among the Origin, Reversed, and Fusion groups. The Tukey post hoc test is applied to correct for multiple comparisons. The results indicate that, for AUROC, there is no significant difference between the Origin and Reversed groups (p = 0.176). However, the Fusion group exhibited significantly higher AUROC values compared to both the Origin (p<0.0001) and Reversed (p<0.0001) groups.

Similarly, in terms of AUPRC, no significant difference is found between the Origin and Reversed groups (p = 0.278), whereas the Fusion group shows significantly improved AUPRC scores relative to the other two groups (p<0.0001). These findings suggest that the proposed TRGC algorithm significantly enhances the AUROC and AUPRC of KANGCI on the fMRI dataset, thereby providing further evidence of its efficacy.

## B.2 In-Depth analysis of the TRGC algorithm

To further clarify the source of the benefits of our proposed method, we design ablation experiments to disentangle the contributions of the KAN and the TRGC algorithm. Specifically, we conduct two additional experiments:

1. **Replacing KAN with a classical MLP in KANGCI**: We substitute KAN in our model with a standard MLP architecture while keeping the TRGC mechanism unchanged.

2. **Modifying the cMLP baseline** : We incorporate the TRGC strategy into the cMLP implementation by Tank et al. (The MLP in cMLP is realized through 1D convolutional layers, which is slightly different from the fully connected MLP used in our setup).

These settings allow us to examine whether the observed performance improvements originate from the structural advantages of KAN or from the fusion strategy itself. The results are provided in Table 8, 9, 10.

Table 8: Overall Performance of KANGCI on Benchmark Datasets

| Models | AUROC | | |
| --- | --- | --- | --- |
| | Origin | Time-Reversed | Fusion |
| Lorenz-96 | 0.983 | 0.895 | 0.986 (↑) |
| fMRI BOLD | 0.685 | 0.629 | 0.685 |
| Dream-3 | 0.648 | 0.547 | 0.652 (↑) |
| Dream-4 | 0.652 | 0.539 | 0.656 (↑) |

Table 9: Replacing KAN with MLP under the Same TRGC algorithm

| Models | AUROC | | |
| --- | --- | --- | --- |
| | Origin | Time-Reversed | Fusion |
| Lorenz-96 | 0.987 | 0.955 | 0.987 |
| fMRI BOLD | 0.724 | 0.719 | 0.745 (↑) |
| Dream-3 | 0.675 | 0.654 | 0.691 (↑) |
| Dream-4 | 0.689 | 0.675 | 0.704 (↑) |

Table 10: Incorporating the TRGC algorithm into cMLP

| Models | AUROC | | |
| --- | --- | --- | --- |
| | Origin | Time-Reversed | Fusion |
| Lorenz-96 | 0.983 | 0.895 | 0.986 (↑) |
| fMRI BOLD | 0.685 | 0.629 | 0.685 |
| Dream-3 | 0.648 | 0.547 | 0.652 (↑) |
| Dream-4 | 0.652 | 0.539 | 0.656 (↑) |

These results demonstrate that TRGC consistently improves performance even when applied to models with classical MLPs, confirming its general effectiveness.

Notably, the performance gain of TRGC on cMLP is modest. We attribute this to cMLP's poor performance on reversed time series. For instance, in the Dream-3 dataset, cMLP achieves only 0.547 AUROC on reversed time series (near random), which limits the benefit of fusion. In contrast, KANGCI maintains stable and balanced performance on both original (0.728) and reversed (0.706) time series, enabling the TRGC mechanism to yield a significant improvement (0.758).

In summary, the performance enhancement of KANGCI arises from the synergistic combination of KAN and TRGC. TRGC can improve inference performance, while KAN's stability across origin and reversed time series enables the fusion to be maximally effective.

## C Experiment on real-world EEG signals

The experiment in Section 4 shows that the proposed model can effectively infer Granger causality from time series. However, these experiments are conducted on simulated datasets, and the applica-

bility and effectiveness of the model on real-world data still need to be further validated. Therefore, in this section, we aim to verify the effectiveness of KANGCI on real-world EEG signals.

## C.1 DATA COLLECTION AND PREPROCESSING

We utilize the EEG dataset provided by Pagnotta et al. (2018a), which comprises somatosensory evoked potentials (SEPs) induced by whisker stimulation of 10 Wistar rats. These rats are anesthetized and subjected to unilateral whisker stimulation via a solenoid for 500 ms across 100 trials. SEPs are recorded using a stainless steel electrode grid positioned on the skulls of the rats, with nodes 1-7 representing the ipsilateral electrodes to stimulation and nodes 9-15 representing the contralateral electrodes to stimulation. SEP signals are sampled at 2000 Hz using a bandpass filtered between 1-500 Hz. The signals contain a time of -100 ms pre- to 200 ms post-stimulation. The analysis pipeline is illustrated in Fig.3.

For the EEG prepossessing, we apply two criteria to identify and exclude trials potentially affected by artifacts. Specifically, a trial is considered contaminated if it meets one of two conditions: (1) the signal variance is higher in the pre-stimulation than in the post-stimulation for over three channels; (2) the signal during the pre-stimulation period exceeds a threshold of 200 $\mu$V in at least one channel (Plomp et al., 2014; Barnett & Seth, 2011). Furthermore, we do not apply any additional filter, as prior research has indicated that filters could compromise the integrity of the informational content and order of data, subsequently affecting the inference of Granger causality (Pullon et al., 2020).

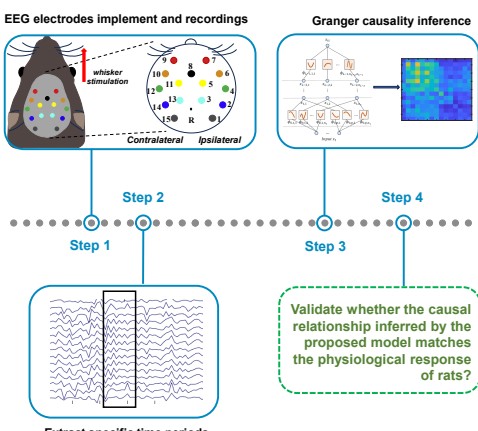

Figure 3: The analysis pipeline of real-world EEG signals. Step 1: EEG electrode positions. Step 2: Extracting the time period to be analyzed. Step 3: Inferring the Granger causality from time series using KANGCI. Step 4: Validating whether the inferred Granger causality matches the physiological response of rats.

## C.2 EVALUATION CRITERIA

We evaluate the performance of KANGCI based on three previously proposed criteria (criteria 2-4) (Plomp et al., 2014; Pagnotta et al., 2018b) and two additional criteria (criteria 1, 5), which collectively examine five distinct characteristics anticipated in the cortical network comprising 15 nodes.

1. **Information flow loss induced by anesthesia**: Information flow between brain regions is essential for sustaining awake consciousness, and anesthesia would induce the loss of information flow (causal relationship), leading to loss of consciousness (Pullon et al., 2020). Consequently, the first criterion is to assess whether the model can detect the absence of Granger causality during the -100 to 0 ms epoch of anesthesia.

2. **Latency differences in sensory cortices**: Stimulation on rat whiskers would activate the primary sensory cortex (S1). However, the latencies of the ipsilateral S1 (iS1, node 4) and contralateral S1 (cS1, node 12) are different (cS1 is about 14ms, iS1 is about 26ms). Therefore, whether the model can infer Granger causality originated from the cS1 and iS1 regions during 10-20ms and 20-30ms, respectively, is the gold standard for evaluating the effectiveness of the model.

3. **Causal driving identification**: Does the model identify the cS1 and iS1 as the main causal driving in the corresponding time period?

4. **Causality from cS1 to contralateral Regions**: Does the model accurately identify the Granger causality from cS1 to the contralateral frontal (node 10) and parietal (node 14) regions?

5. **Causality from iS1 to ipsilateral Regions**: Does the model accurately identify the Granger causality from iS1 to the ipsilateral frontal (node 2) and parietal (node 6) regions?

## C.3 RESULTS

As shown in Fig.4(a), KANGCI does not detect any significant causal relationship from -100 to 0 milliseconds pre-stimulation, and the causal driving of each channel is only around 0.06 (Fig.4(b)). These results indicate that the model detects the absence (loss) of Granger causality caused by anesthesia during the non-stimulation period (criterion 1).

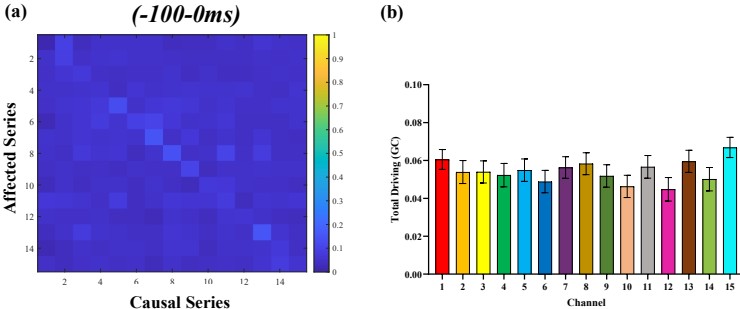

Figure 4: (a) The inferred Granger causality in -100-0 ms. (b) The Granger causality driving of each channel.

Furthermore, KANGCI effectively identifies the causal relationship from cS1 to the contralateral frontal and parietal regions during the 10-20 ms epoch, as illustrated in Fig.5(a). Meanwhile, we conduct statistical analysis of the causal driving for all channels using one-way ANOVA (Fig.5(b)). The result shows that the causal driving of cS1 is significantly greater than that of all other nodes ($p<0.0001$), indicating that cS1 is the primary causal driver during 10-20 ms. These findings satisfy evaluation criteria 2, 3, and 4.

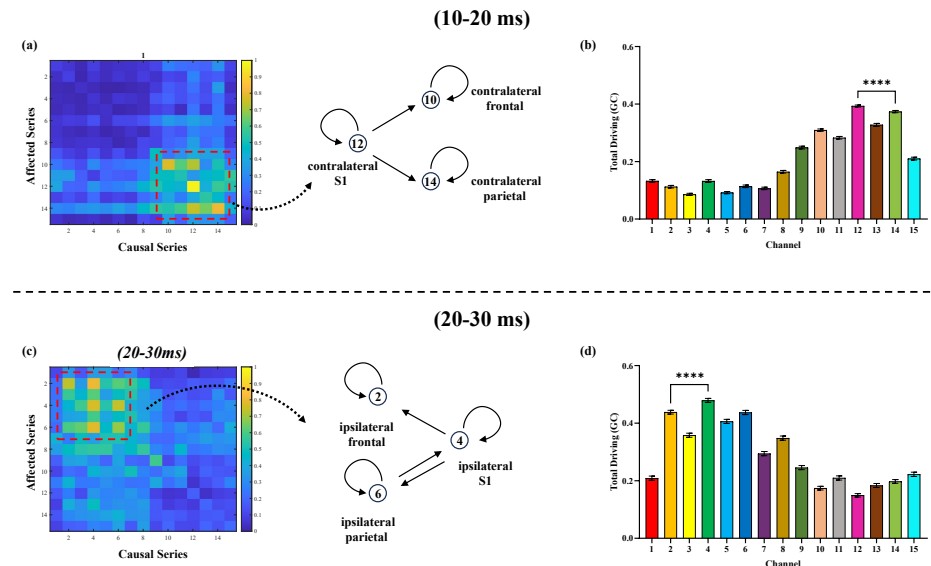

Figure 5: (a) The inferred Granger causality in 10-20 ms. (b) The causal driving of each channel in 10-20 ms. (c) The inferred Granger causality in 20-30 ms. (b) The causal driving of each channel in 20-30 ms.

During the time period of 20-30 ms, KANGCI successfully identifies the causal relationship from iS1 to the ipsilateral frontal and parietal regions, as depicted in Fig.5(c). One-way ANOVA also reveals that the causal driving of iS1 is significantly larger than that of all other nodes ($p<0.0001$) (Fig.5(d)), indicating that iS1 is the primary causal driving during 20-30 ms. Consequently, these results match evaluation criteria 2, 3, and 5.

Therefore, these findings collectively highlight KANGCI's efficiency in identifying distinct causal relationships across various time periods, validating KANGCI's ability to infer Granger causality from real-world EEG signals.

## D  AUPRC: AN ESSENTIAL METRIC FOR ASSESSING MODEL PERFORMANCE IN INFERRING SPARSE CAUSAL RELATIONSHIPS

Sparse causality characterizes scenarios where only a small subset of variables exhibit true causal relationships while the majority do not. For example, the ground-truth causal matrices in the fMRI (sim-4) and DREAM3 (Ecoli-1) datasets contain only 2.44% and 1.25% true causal relationships, respectively. In such sparse causal inference scenarios, AUPRC offers a more informative evaluation metric than AUROC, as it better reflects the model's capacity to identify sparse causal relationships. Consequently, AUPRC is considered a critical metric for assessing model performance in sparse causal inference tasks.

Table 11: AUPRC of the fMRI BOLD signals, Subject=50, T=50/100/200/2000/5000

| Dateset | AUPRC | | | | | | | | | | |
|---|---|---|---|---|---|---|---|---|---|---|---|
| | cMLP | cLSTM | TCDF | eSRU | GVAR | NAVAR (MLP) | NAVAR (LSTM) | JGC | CUTS+ | JRNGC | **KANGCI** |
| Sim1 | 0.634±0.04 | 0.655±0.05 | 0.728±0.03 | 0.703±0.04 | 0.716±0.05 | 0.644±0.03 | 0.673±0.05 | 0.653±0.05 | 0.753±0.04 | 0.772±0.04 | **0.784**±0.08 |
| Sim2 | 0.619±0.05 | 0.643±0.04 | 0.714±0.04 | 0.688±0.04 | 0.705±0.04 | 0.632±0.03 | 0.655±0.03 | 0.734±0.02 | 0.711±0.03 | 0.729±0.03 | **0.744**±0.03 |
| Sim3 | 0.626±0.06 | 0.627±0.05 | 0.702±0.03 | 0.669±0.04 | 0.681±0.05 | 0.606±0.03 | 0.637±0.04 | 0.723±0.02 | 0.719±0.02 | **0.745**±0.03 | 0.723±0.02 |
| Sim4 | 0.540±0.06 | 0.595±0.05 | 0.638±0.03 | 0.613±0.04 | 0.637±0.04 | 0.554±0.04 | 0.543±0.05 | 0.617±0.02 | 0.662±0.02 | 0.651±0.01 | **0.674**±0.01 |
| Sim5 | 0.702±0.05 | 0.739±0.04 | 0.771±0.04 | 0.767±0.04 | 0.759±0.03 | 0.741±0.03 | 0.743±0.04 | 0.756±0.03 | 0.774±0.04 | 0.781±0.05 | **0.809**±0.05 |
| Sim6 | 0.662±0.15 | 0.683±0.09 | 0.740±0.02 | 0.721±0.04 | 0.746±0.03 | 0.722±0.03 | 0.746±0.03 | 0.759±0.03 | 0.781±0.03 | 0.774±0.03 | **0.806**±0.02 |
| Sim7 | 0.686±0.05 | 0.671±0.04 | 0.756±0.04 | 0.742±0.04 | 0.719±0.03 | 0.773±0.03 | 0.789±0.03 | 0.800±0.03 | 0.826±0.05 | 0.818±0.04 | **0.856**±0.04 |
| Sim8 | 0.467±0.15 | 0.439±0.09 | 0.637±0.08 | 0.583±0.09 | 0.601±0.07 | 0.574±0.12 | 0.557±0.11 | 0.592±0.09 | 0.650±0.08 | 0.674±0.07 | **0.725**±0.08 |
| Sim9 | 0.648±0.07 | 0.652±0.09 | 0.743±0.06 | 0.693±0.05 | 0.659±0.06 | 0.685±0.08 | 0.701±0.08 | 0.739±0.07 | 0.781±0.06 | 0.778±0.06 | **0.794**±0.08 |
| Sim10 | 0.604±0.07 | 0.619±0.09 | 0.705±0.04 | 0.630±0.11 | 0.647±0.08 | 0.641±0.11 | 0.693±0.12 | 0.652±0.08 | **0.768**±0.07 | 0.736±0.08 | 0.745±0.07 |
| Sim11 | 0.661±0.04 | 0.655±0.03 | 0.727±0.03 | 0.693±0.04 | 0.682±0.03 | 0.703±0.03 | 0.712±0.03 | 0.749±0.03 | **0.785**±0.02 | 0.766±0.03 | 0.771±0.03 |
| Sim12 | 0.688±0.05 | 0.676±0.03 | 0.740±0.04 | 0.682±0.03 | 0.671±0.04 | 0.694±0.03 | 0.706±0.03 | 0.722±0.05 | 0.738±0.04 | 0.753±0.04 | **0.778**±0.03 |
| Sim13 | 0.537±0.07 | 0.529±0.04 | 0.688±0.06 | 0.609±0.08 | 0.631±0.09 | 0.654±0.08 | 0.672±0.09 | 0.659±0.09 | 0.685±0.07 | 0.712±0.07 | **0.782**±0.08 |
| Sim14 | 0.626±0.08 | 0.632±0.07 | 0.707±0.06 | 0.663±0.07 | 0.654±0.09 | 0.680±0.08 | 0.687±0.07 | 0.701±0.06 | 0.737±0.07 | 0.724±0.06 | **0.746**±0.08 |
| Sim15 | 0.608±0.10 | 0.613±0.09 | 0.649±0.06 | 0.574±0.09 | 0.596±0.08 | 0.648±0.07 | 0.641±0.09 | 0.656±0.08 | 0.673±0.08 | **0.735**±0.09 | 0.699±0.08 |
| Sim16 | 0.576±0.11 | 0.588±0.13 | 0.676±0.08 | 0.624±0.09 | 0.604±0.09 | 0.598±0.07 | 0.624±0.09 | 0.616±0.12 | 0.711±0.09 | 0.705±0.11 | **0.748**±0.09 |
| Sim17 | 0.647±0.05 | 0.634±0.05 | 0.755±0.04 | 0.680±0.04 | 0.675±0.05 | 0.705±0.04 | 0.727±0.04 | 0.731±0.04 | 0.753±0.03 | **0.802**±0.04 | 0.777±0.03 |
| Sim18 | 0.615±0.07 | 0.623±0.07 | 0.748±0.03 | 0.649±0.05 | 0.662±0.06 | 0.685±0.06 | 0.713±0.05 | 0.720±0.06 | 0.786±0.05 | 0.773±0.05 | **0.797**±0.06 |
| Sim19 | 0.749±0.05 | 0.751±0.04 | 0.786±0.03 | 0.744±0.05 | 0.702±0.06 | 0.763±0.04 | 0.801±0.04 | 0.825±0.04 | 0.849±0.03 | 0.831±0.03 | **0.872**±0.03 |
| Sim20 | 0.743±0.04 | 0.786±0.09 | 0.837±0.02 | 0.811±0.03 | 0.769±0.05 | 0.805±0.03 | 0.833±0.04 | 0.842±0.02 | 0.879±0.03 | 0.861±0.02 | **0.885**±0.03 |
| Sim21 | 0.629±0.07 | 0.638±0.08 | 0.727±0.05 | 0.674±0.06 | 0.681±0.04 | 0.653±0.05 | 0.677±0.06 | 0.612±0.08 | 0.752±0.06 | 0.739±0.06 | **0.770**±0.07 |
| Sim22 | 0.656±0.06 | 0.643±0.06 | 0.711±0.05 | 0.685±0.07 | 0.698±0.05 | 0.628±0.07 | 0.650±0.06 | 0.649±0.07 | 0.753±0.05 | 0.781±0.06 | **0.804**±0.06 |
| Sim23 | 0.574±0.08 | 0.598±0.09 | 0.573±0.05 | 0.568±0.08 | 0.557±0.09 | 0.549±0.09 | 0.551±0.08 | 0.564±0.09 | 0.602±0.08 | **0.617**±0.09 | 0.584±0.08 |
| Sim24 | 0.471±0.09 | 0.505±0.13 | 0.536±0.06 | 0.529±0.06 | 0.523±0.04 | 0.468±0.11 | 0.516±0.12 | 0.511±0.07 | **0.546**±0.07 | 0.539±0.07 | 0.522±0.09 |
| Sim25 | 0.602±0.07 | 0.580±0.05 | 0.657±0.04 | 0.619±0.07 | 0.623±0.05 | 0.588±0.05 | 0.574±0.07 | 0.628±0.04 | 0.695±0.06 | 0.717±0.06 | **0.732**±0.08 |
| Sim26 | 0.559±0.07 | 0.551±0.07 | 0.632±0.07 | 0.595±0.06 | 0.606±0.07 | 0.577±0.06 | 0.547±0.06 | 0.611±0.05 | 0.666±0.08 | 0.671±0.07 | **0.685**±0.09 |
| Sim27 | 0.614±0.08 | 0.605±0.06 | 0.663±0.06 | 0.628±0.09 | 0.670±0.06 | 0.603±0.07 | 0.571±0.06 | 0.638±0.06 | 0.694±0.07 | 0.706±0.06 | **0.712**±0.08 |
| Sim28 | 0.641±0.06 | 0.635±0.05 | 0.733±0.04 | 0.685±0.06 | 0.703±0.05 | 0.619±0.04 | 0.582±0.04 | 0.718±0.07 | 0.732±0.08 | 0.758±0.06 | **0.796**±0.07 |

Table 12: AUPRC of Dream-3, T=966, p=100

| Models | AUROC | | | | |
|---|---|---|---|---|---|
| | Ecoli-1 | Ecoli-2 | Yeast-1 | Yeast-2 | Yeast-3 |
| cMLP | 0.023 | 0.019 | 0.020 | 0.015 | 0.014 |
| cLSTM | 0.017 | 0.017 | 0.015 | 0.023 | 0.031 |
| TCDF | 0.012 | 0.011 | 0.014 | 0.014 | 0.013 |
| eSRU | 0.036 | 0.034 | 0.041 | 0.052 | 0.044 |
| GVAR | 0.103 | 0.117 | 0.098 | 0.103 | 0.104 |
| NAVAR (MLP) | 0.102 | 0.107 | 0.073 | 0.105 | 0.089 |
| NAVAR (LSTM) | 0.013 | 0.012 | 0.030 | 0.038 | 0.052 |
| JGC | 0.018 | 0.016 | 0.026 | 0.050 | 0.059 |
| CUTS+ | 0.154 | 0.143 | 0.121 | 0.128 | 0.105 |
| JRNGC | 0.198 | **0.202** | **0.172** | 0.142 | 0.130 |
| **KANGCI** | **0.207** | 0.163 | 0.154 | **0.147** | **0.132** |

Table 13: AUPRC of Dream-4, T=210, p=100

| Models | AUROC | | | | |
|---|---|---|---|---|---|
| | Gene-1 | Gene-2 | Gene-3 | Gene-4 | Gene-5 |
| cMLP | 0.017 | 0.015 | 0.022 | 0.025 | 0.018 |
| cLSTM | 0.035 | 0.044 | 0.037 | 0.041 | 0.033 |
| TCDF | 0.011 | 0.013 | 0.014 | 0.012 | 0.014 |
| eSRU | 0.045 | 0.041 | 0.043 | 0.051 | 0.059 |
| GVAR | 0.103 | 0.094 | 0.082 | 0.077 | 0.065 |
| NAVAR (MLP) | 0.095 | 0.113 | 0.096 | 0.085 | 0.077 |
| NAVAR (LSTM) | 0.014 | 0.017 | 0.023 | 0.026 | 0.015 |
| JGC | 0.035 | 0.028 | 0.031 | 0.037 | 0.025 |
| CUTS+ | 0.054 | 0.044 | 0.041 | 0.057 | 0.051 |
| JRNGC | 0.165 | 0.146 | 0.134 | **0.129** | **0.117** |
| **KANGCI** | **0.181** | **0.167** | **0.146** | 0.122 | 0.105 |

# E   VISUALIZATION OF THE B-SPLINE FUNCTION DURING THE CAUSAL INFERENCE

Interpretability is a key advantage of KAN compared to other causal inference models. Different from MLPs, which use fixed weight matrices, KAN replaces these fixed weights with learnable univariate functions assigned to each edge in the network. This design enables the model to be easily visualized and interpreted. In our study, we conducted a preliminary interpretability analysis by visualizing B-spline basis functions and their corresponding control point coefficients, as shown in Appendix E. These visualizations provide insights into how the model captures causal relationships by adaptively shaping the basis functions through learning —a capability not available in prior causal models, such as cMLP/cLSTM. We use the Lorenz-96 dataset, KANGCI achieves its peak performance at approximately 80 training epochs, maintaining stability thereafter. The corresponding results are illustrated in Fig. 6.

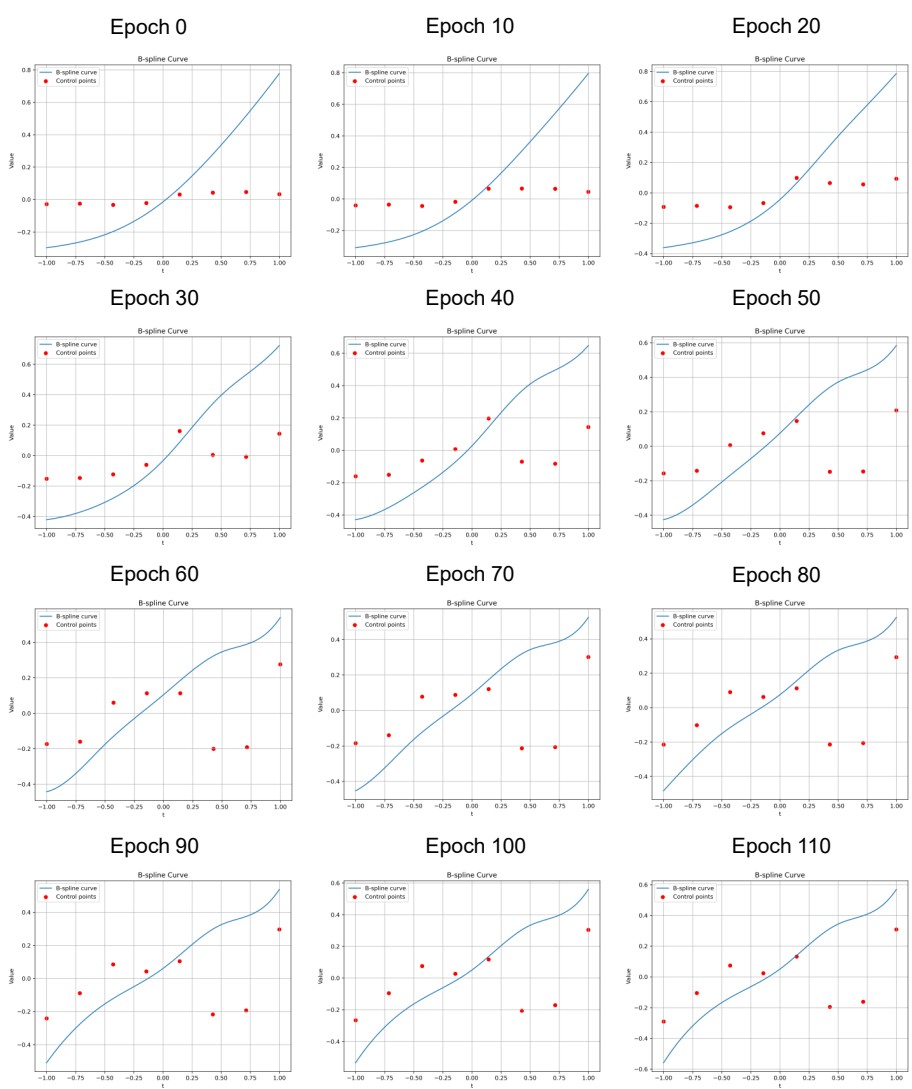

Figure 6: Visualization of the B-spline function and the corresponding control point coefficients during the causal inference.

## F    SENSITIVITY AGAINST HYPERPARAMETER TUNING

We conduct experiments on the Lorenz-96, DREAM3, DREAM4, and fMRI datasets to evaluate the robustness of the model with respect to hyperparameter variations. The tuning strategy is group lasso hyperparameter $\lambda \in [1e^{-3}, 1e^{-1}]$, ridge regularization $\gamma \in [1, 20]$. The results are shown as follows:

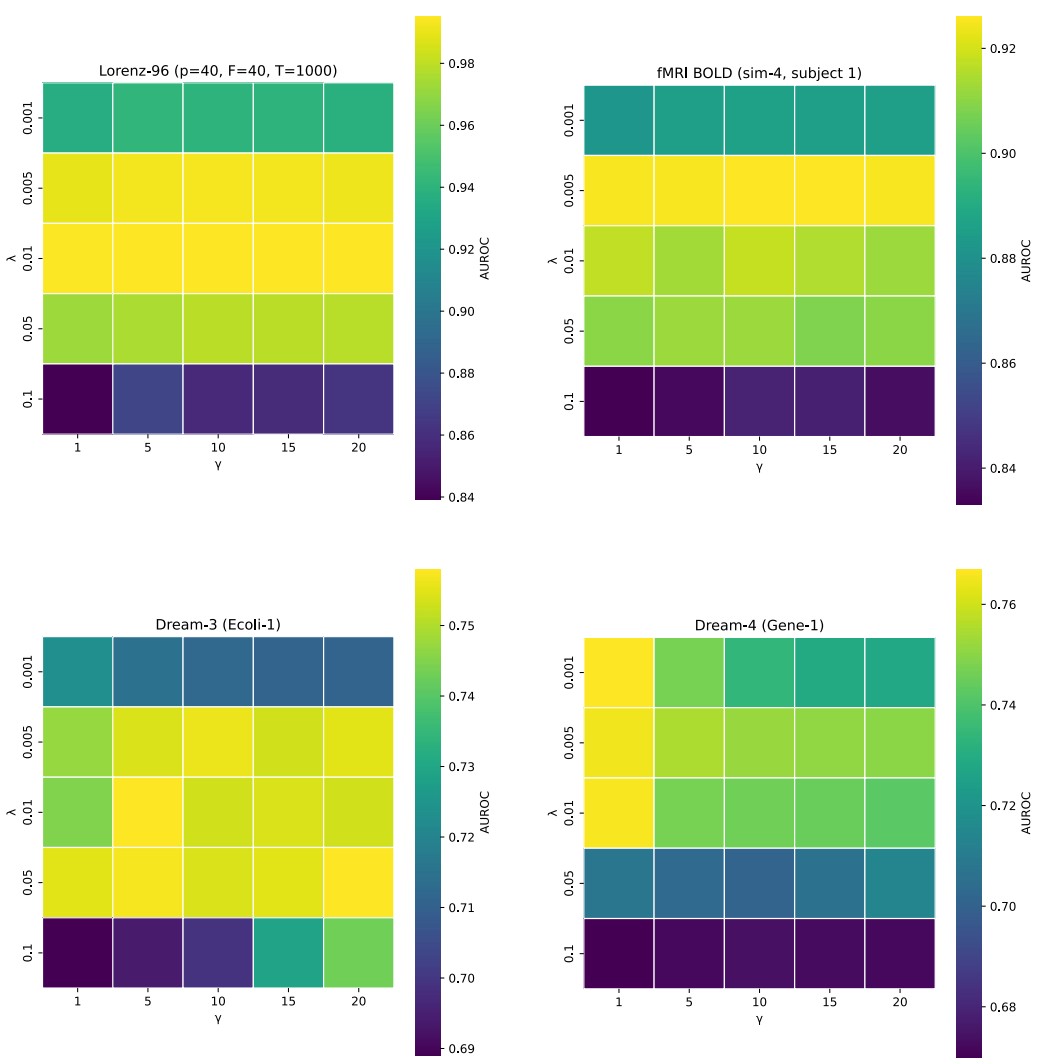

Figure 7: KANGCI hyperparameter tuning results.

The results demonstrates that the model consistently maintains stable performance across a range of hyperparameter settings, thereby validating the robustness of KANGCI.

## G    COMPARISON WITH BASIS FUNCTION-BASED GRANGER CAUSALITY MODELS

As the KANGCI, in effect, is a universal basis function model, it is essential to compare its performance with other basis function-based Granger causality models. Accordingly, we present a comparative analysis of KANGCI against MLCausality (Fulmyk et al., 2023) and NonlinCausalityNN (Rosoł et al., 2022) across all datasets considered in Section 4.

Table 14: Comparison of AUROC for basis function-based Granger causality models.

| | KANGCI | MLCausality | NonlinCausalityNN |
|---|---|---|---|
| Lorenz-96 ($F = 40$, $p = 40$, $T = 1000$) | **0.995**$_{\pm 0.002}$ | 0.779$_{\pm 0.051}$ | 0.835$_{\pm 0.044}$ |
| fMRI BOLD (all simulations) | **0.815**$_{\pm 0.078}$ | 0.592$_{\pm 0.042}$ | 0.639$_{\pm 0.063}$ |
| Dream-3 | **0.643**$_{\pm 0.086}$ | 0.509$_{\pm 0.018}$ | 0.524$_{\pm 0.023}$ |
| Dream-4 | **0.631**$_{\pm 0.065}$ | 0.505$_{\pm 0.009}$ | 0.533$_{\pm 0.027}$ |
| VAR ($sparsity = 0.2$, $lag = 3$) | **1.000**$_{\pm 0.000}$ | 0.902$_{\pm 0.006}$ | 0.937$_{\pm 0.004}$ |
| Basis function | B-spline basis function | Radial basis function | Generalized radial basis function |

As illustrated in Table 14, both MLCausality and NonlinCausalityNN demonstrate effective causal inference performance in the linear VAR dataset. However, their effectiveness declines markedly in nonlinear datasets. In comparison, KANGCI demonstrates superior performance across all datasets, consistently outperforming other basis function-based causal models.

In addition, we compared KANGCI with the contemporaneous work GC-KAN (Lin et al., 2024). As the implementation of GC-KAN is not publicly available, the comparison is limited to the results (Lorenz-96) reported in their publication, as presented in Table 15.

Table 15: Comparison of AUROC for other KAN-based causal model.

| | KANGCI | GC-KAN |
|---|---|---|
| Lorenz-96 ($F = 40$, $p = 20$, $T = 500$) | **0.969**$_{\pm 0.1}$ | 0.871$_{\pm 0.4}$ |
| Lorenz-96 ($F = 40$, $p = 20$, $T = 1000$) | **1.000**$_{\pm 0.0}$ | 0.957$_{\pm 0.2}$ |
| Basis function | B-spline basis function | B-spline basis function |

Specifically, Lin et al. (2024) reported that GC-KAN achieved an AUROC of 0.957 on the Lorenz-96 dataset ($F = 40$, $p = 20$, $T = 1000$). However, its performance dropped to 0.871 when the time series length was reduced to $T = 500$, even lower than that of cMLP (AUROC = 0.896) under the same conditions.

## H    HYPERPARAMETERS AND CONFIGURATIONS

Table 16, 17, 18, 19, 20 illustrate the configurations and hyperparameters of KANGCI used for comparison. For the comparative models, we adopt the best-performing hyperparameters provided by Khanna & Tan (2019) and Zhou et al. (2024), which encompassed the hyperparameters for all tested models.

Table 16: The detailed parameters and configurations of the Lorenz-96 dataset.

| | Hidden size units | Group Lasso | Ridge regularization | learning rate |
|---|---|---|---|---|
| Lorenz-96 ($F = 10$, $P = 10$, $T = 1000$) | 10 | 0.01 | 20 | 0.001 |
| Lorenz-96 ($F = 40$, $P = 40$, $T = 1000$) | 20 | 0.01 | 5 | 0.001 |
| Lorenz-96 ($F = 40$, $P = 40$, $T = 500$) | 20 | 0.01 | 5 | 0.001 |

## I    THE SIMULATIONS' SPECIFICATION OF THE FMRI BOLD DATASET

The simulations' specification are shown in the Table 21.

Table 17: The detailed parameters and configurations of the fMRI BOLD dataset.

|  | Hidden size units | Group Lasso | Ridge regularization | learning rate |
|---|---|---|---|---|
| fMRI (all the simulations) | [128,64,32] | 0.01 | 5 | 0.001 |

Table 18: The detailed parameters and configurations of the Dream-3 dataset.

|  | Hidden size units | Group Lasso | Ridge regularization | learning rate |
|---|---|---|---|---|
| Ecoli-1 | 128 | 0.01 | 5 | 0.01 |
| Ecoli-2 | 128 | 0.01 | 5 | 0.01 |
| Yeast-1 | 128 | 0.01 | 5 | 0.01 |
| Yeast-2 | 128 | 0.01 | 5 | 0.01 |
| Yeast-3 | 128 | 0.01 | 5 | 0.01 |

Table 19: The detailed parameters and configurations of the Dream-4 dataset.

|  | Hidden size units | Group Lasso | Ridge regularization | learning rate |
|---|---|---|---|---|
| Gene-1 | 128 | 0.0001 | 5 | 0.01 |
| Gene-2 | 128 | 0.0001 | 5 | 0.01 |
| Gene-3 | 128 | 0.0001 | 5 | 0.01 |
| Gene-4 | 128 | 0.0001 | 5 | 0.01 |
| Gene-5 | 64 | 0.0005 | 5 | 0.01 |

Table 20: The detailed parameters and configurations of the VAR dataset.

|  | Hidden size units | Group Lasso | Ridge regularization | learning rate |
|---|---|---|---|---|
| VAR | 16 | 0.0001 | 0.05 | 0.01 |

Table 21: The simulations' specification of fMRI BOLD dataset

| Dataset | Dimension | Time Samples | Noise | HRF std.dev. (s) | Other descriptions |
|---|---|---|---|---|---|
| Sim1 | 5 | 200 | 1% | 0.5 | |
| Sim2 | 10 | 200 | 1% | 0.5 | |
| Sim3 | 15 | 200 | 1% | 0.5 | |
| Sim4 | 50 | 200 | 1% | 0.5 | |
| Sim5 | 5 | 1200 | 1% | 0.5 | |
| Sim6 | 10 | 1200 | 1% | 0.5 | |
| Sim7 | 5 | 5000 | 1% | 0.5 | |
| Sim8 | 5 | 200 | 1% | 0.5 | Shared inputs |
| Sim9 | 5 | 5000 | 1% | 0.5 | Shared inputs |
| Sim10 | 5 | 200 | 1% | 0.5 | Global mean confound |
| Sim11 | 10 | 200 | 1% | 0.5 | Bad ROIs (Time series mixed with each other) |
| Sim12 | 10 | 200 | 1% | 0.5 | Bad ROIs (Mixed addition time series) |
| Sim13 | 5 | 200 | 1% | 0.5 | Existing backwards connections |
| Sim14 | 5 | 200 | 1% | 0.5 | Existing cyclic connections |
| Sim15 | 5 | 200 | 0.1% | 0.5 | Connection strength becomes stronger |
| Sim16 | 5 | 200 | 1% | 0.5 | Existing more connections |
| Sim17 | 10 | 200 | 0.1% | 0.5 | |
| Sim18 | 5 | 200 | 1% | - | |
| Sim19 | 5 | 2400 | 0.1% | 0.5 | Time lag=100ms |
| Sim20 | 5 | 2400 | 0.1% | - | Time lag=100ms |
| Sim21 | 5 | 200 | 1% | 0.5 | 2-group test |
| Sim22 | 5 | 200 | 0.1% | 0.5 | Connection strength is non stationary |
| Sim23 | 5 | 200 | 0.1% | 0.5 | Connection strength is stationary |
| Sim24 | 5 | 200 | 0.1% | 0.5 | Only one strong external input |
| Sim25 | 5 | 100 | 1% | 0.5 | |
| Sim26 | 5 | 50 | 1% | 0.5 | |
| Sim27 | 5 | 50 | 0.1% | 0.5 | |
| Sim28 | 5 | 100 | 0.1% | 0.5 | |

## J   THE USE OF LARGE LANGUAGE MDOLES (LLMs)

In this work, LLMs were used solely to aid in polishing the language and improving the clarity of writing. All research ideas, methodological designs, experiments, and analyses were conceived and conducted by the authors. The authors take full responsibility for the content of this paper.

