# OpenReview forum: "Kolmogorov-Arnold Networks for Time Series Granger Causality Inference"
_ICLR.cc/2026/Conference — ICLR 2026 Conference Withdrawn Submission_

### Official Review · Reviewer_HNWF · 2025-10-26

**Soundness:** 3
**Presentation:** 3
**Contribution:** 3
**Rating:** 6
**Confidence:** 4

**Summary:**

KANGCI adapts Kolmogorov-Arnold Networks (KANs) to nonlinear Granger causality inference.

**Strengths:**

1. Application of KAN to causal inference; theoretically grounded via KA representation.
2. Integrates sparsity control directly into functional decomposition.
3. Extensive benchmark coverage from synthetic to real data.

**Weaknesses:**

1. Presentation dense; many equations without explanatory intuition.
2. Comparison baselines could include more recent nonlinear GC methods (e.g., NeuralODE-GC).
3. Unclear how spline control points affect interpretability and regularization strength.
4. Limited discussion on computational complexity of KAN layers.

**Questions:**

1. It would be valuable to include an ablation study to evaluate the contribution of ridge and sparsity penalties in your model.
2. Please provide more detailed information on the hyperparameter tuning process and how the model’s training stability is ensured. Specifically, which hyperparameters were tuned, and what criteria were used to select the best values? Additionally, a discussion of any challenges faced during training, such as overfitting, convergence issues, or sensitivity to hyperparameter changes, would be beneficial.
3. While the model shows promising results for Granger causality (GC), it would be important to discuss its generalizability to multivariate nonlinear interventions. How does KANGCI handle more complex interactions, such as those found in systems with nonlinearities or multiple simultaneous interventions? Does the model still perform effectively under these conditions, and if so, what modifications or considerations would be necessary to apply it to such cases?
4. To enhance the interpretability and intuition of your model, it would be helpful to include visualizations of the inferred causal graphs. This can allow readers to better understand the relationships between variables and the causal structure your model identifies.

---

### Official Review · Reviewer_Tvbi · 2025-10-31

**Soundness:** 2
**Presentation:** 2
**Contribution:** 2
**Rating:** 2
**Confidence:** 3

**Summary:**

This is a paper of interest to researchers and practitioners doing causal discovery from multivariate time series (e.g., ML for neuroscience, econometrics, and systems biology). It describes an incremental improvement over existing methods and tests it mostly on simulated data.

- Proposes a component-wise neural Granger framework using a Kolmogorov–Arnold Network (KAN) backbone; edges are read from group-sparsified first-layer parameters.
- Introduces a time-reversed “choose-or-fuse” heuristic: train on forward and reversed series; pick the graph with better prediction and sparsity losses, or average them if criteria disagree.
- Evaluates against known ground-truth graphs on simulated Lorenz-type dynamics, gene-network suites, an extensive simulated fMRI suite, and linear VAR; includes a real EEG case study for plausibility.
- Reports small–moderate metric gains in several challenging simulated settings; ties in easier/linear ones; ablations show the fusion heuristic often helps.

**Strengths:**

- Broad and carefully executed simulation coverage with known ground truth (facilitates clear metric comparisons).
- Simple, reproducible pipeline (component-wise models; straightforward sparsity/read-out).
- Empirical robustness of the fusion heuristic in several noisy, sparse regimes.

**Weaknesses:**

- Limited methodological novelty: swapping in KAN plus an ad hoc time-reversal rule; no new identification theory.
- Evidence framing: improvements are mostly modest, not consistently significant, and not capacity/compute-matched against the strongest baselines.
- Interpretability of “control”: time-reversal is positioned as a robustness device, but it is not a principled control outside linear settings; no explicit false-positive calibration (e.g., edge-wise FPR/FDR).

**Questions:**

- Justification for KAN: Why should KAN outperform MLP/RNN under equal parameter or FLOP budgets? Provide controlled sweeps.
- Error calibration: How are thresholds chosen; what is the edge-wise FPR/FDR at matched TPR?
- Compute fairness: Report parameter counts, FLOPs, wall-clock, and VRAM for all methods compared; ensure epoch/optimizer budgets are aligned.
- Sensitivity to lag and nonstationarity: Specify lag selection, mis-specification robustness, and behavior under structural breaks or latent confounding.

---

### Official Review · Reviewer_8nv3 · 2025-10-31

**Soundness:** 3
**Presentation:** 3
**Contribution:** 4
**Rating:** 8
**Confidence:** 4

**Summary:**

The authors modify the KAN neural network architecture to incorporating the sparsity-inducing penalty and
ridge regularization, resulting in a granger causal inference scheme.

**Strengths:**

This is a very nice architecture that has been baselined against some of the leading causal inference methods available.   Interestingly, KAN is often not that good for reconstruction and forecasting tasks, but the modification for inference seems to allow it to perform well for this task.

**Weaknesses:**

The authors don't offer much in terms of how the method holds up under noise and data corruption.  This would seem to be a valuable evaluation that could easily be done on the Lorenz96 model.

**Questions:**

How robust is the method?  Does it actually hold up and beat the competing methods as you increase the noise and corruption of data.  Can the authors perform such a study on, for instance, Lorenz96 data.

---

### Official Review · Reviewer_kiCm · 2025-11-01

**Soundness:** 2
**Presentation:** 2
**Contribution:** 1
**Rating:** 2
**Confidence:** 3

**Summary:**

This paper introduces KANGCI, a novel model that adapts Kolmogorov–Arnold Networks (KAN) for Granger causality inference on nonlinear time series. The authors extract base weights from KAN layers, apply group lasso and ridge regularization, and propose a time-reversed Granger causality (TRGC) algorithm to enhance robustness and mitigate spurious correlations. The method is validated on multiple benchmark datasets and shows competitive or superior AUROC/AUPRC scores compared to existing baselines.

**Strengths:**

1. **Methods:** The paper introduces a creative extension of KAN to the problem of Granger causality inference. By leveraging the Kolmogorov–Arnold representation theorem and spline-based nonlinearities, the proposed framework captures complex, non-smooth, and high-dimensional dependencies that are difficult for standard MLP or RNN models. And the model utilized the TRGC algorithm, which is a simple yet effective strategy that improves robustness and reduces spurious correlations.

2. **Evaluation:** The model is evaluated on a good range of benchmarks, which helps establish that it behaves consistently across different types of nonlinear and noisy time series. The performance is generally competitive with or better than existing baselines.

**Weaknesses:**

1. **Novelty:** Although the paper combines KAN with Granger causality inference, the conceptual leap from prior work—particularly GC-KAN. The main innovations are the use of time-reversal fusion, which may be seen as incremental rather than fundamentally new.

2. **Methods:** The approach still relies on Granger causality, which reflects prediction rather than real cause-and-effect relationships. While the method models nonlinear patterns more accurately, it does not solve core issues like confounding variables or lack of interventions.

3. **Interpretation:** The paper invokes the Kolmogorov–Arnold theorem to motivate KAN but does not connect it to causal identifiability or provide intuition for why this representation improves inference accuracy.

4. **Writing:** The manuscript sometimes reads as a technical report rather than a research contribution.Many sections simply report formulations and performance numbers without offering deeper discussion or interpretation.

**Questions:**

1. Could the authors clearly explain how KANGCI differs from GC-KAN and other recent neural Granger causality models? Specifically, what unique modeling or algorithmic aspect makes KANGCI more than a combination of KAN and TRGC?

2. Discuss the fundamental gaps and limitations between the predictive influence of the models between true causation. Any plans to extend the model toward structural or interventional causal frameworks?

3.  Provide Details on hyperparameter selection, and it does impact the results?

4. Improve the writing, since the current paper is technically detailed but lacks intuitive explanations of design choices.

---

### Note · Authors · 2025-12-18

I have read and agree with the venue's withdrawal policy on behalf of myself and my co-authors.